# Regional and cell specific bioactivity of injectable extracellular matrix biomaterials in myocardial infarction

Joshua M. Mesfin [1,2], Van K. Ninh [3], Miranda D. Diaz[1,2], Michael B. Nguyen[1,2], Alexander Chen [1,2,4], Raymond M. Wang[1,2], Elyse G. Wong[1,2], Maria L. Karkanitsa [1,2], Jervaughn D. Hunter[1,2], Justin Yu[3], Benjamin D. Bridgelal[1,2], John-Paul A. Pham[1,2], Nika Taghdiri[1], David M. Calcagno[1], Colin G. Luo [2,3], Rebecca L. Braden[1,2], Zhenxing Fu[3], Kevin R. King[1,3] ✉ & Karen L. Christman [1,2,4,5] ✉

Myocardial infarction (MI) remains a global health concern. To mitigate subacute and chronic MI pathophysiology, we previously investigated a pro-reparative decellularized extracellular matrix hydrogel. Despite increasing interest in biomaterial scaffolds, single cell and spatially resolved transcriptomics have not been used to probe their therapeutic activity in the heart. Here, we utilize spatial transcriptomics and single nucleus RNA sequencing to delineate the regional and cell-specific bioactivity of extracellular matrix biomaterials. Extracellular matrix hydrogel subacute treatment in female rats induces cardiac resident macrophage preservation, fibroblast activation, and increased lymphatic, vasculature, smooth muscle, and cardiomyocyte development as well as neurogenesis. Chronic treatment in female rats elicits macrophage polarization, neurogenesis, and development of cardiomyocytes, endothelial cells, and fibroblasts. When comparing treatment timepoints, subacute administration has stronger immune modulation, while chronic administration demonstrates higher cardiac development markers. Both subacute and chronic administration are associated with fibroblast activation and vasculature development. Thus, we elucidate undiscovered therapeutic targets of an injectable extracellular matrix hydrogel, further demonstrating the potential of these biomaterials as an MI therapy.

Ischemic injury due to myocardial infarction (MI) leads to cardiomyocyte apoptosis, immune cell infiltration, loss of vascularization, and fibrosis[1]. After MI, patients often undergo negative left ventricular (LV) remodeling, develop reduced cardiac function, and eventually progress to heart failure[2]. Current MI treatments include percutaneous or surgical revascularization and small-molecule drugs. Experimental pro-regenerative therapies such as gene[3] and cell therapies[4] have also been extensively explored.

As an alternative approach, we and others have utilized injectable biomaterial therapeutics as an acellular and more cost-effective

[1]Shu Chien-Gene Lay Department of Bioengineering, University of California San Diego, La Jolla, CA, USA. [2]Sanford Consortium for Regenerative Medicine, La Jolla, CA, USA. [3]School of Medicine, University of California San Diego, San Diego, CA, USA. [4]Program in Materials Science and Engineering, University of California San Diego, La Jolla, USA. [5]Sanford Stem Cell Institute, University of California San Diego, La Jolla, CA, USA. ✉e-mail: krking@health.ucsd.edu; kchristman@ucsd.edu

strategy[5-7]. We previously developed an injectable biomaterial derived from decellularized porcine myocardium as acellular, pro-regenerative strategy for treating MI. This ECM hydrogel derived from decellularized and digested porcine left ventricular myocardium[8] showed safety and efficacy in small and large animal subacute MI models[9-11], in a small animal chronic MI model[12], and initial safety and feasibility in a phase 1 clinical trial in subacute and chronic MI patients[13]. Through histology, whole transcript arrays, and/or target gene expression analyses, the ECM hydrogel was shown to be immunomodulatory through promotion of a pro-remodeling immune phenotype, increase neovascularization, and decreased cardiomyocyte apoptosis, fibrosis and negative left ventricular remodeling[9,11,12,14].

Coupled with single-nucleus RNA sequencing (snRNAseq), spatial transcriptomics is a tool that has delineated the transcriptomic progression of MI[15-17] and the emergence of the border zone[18-20]. However, to date, these techniques have just recently been used to assess the regional and cell-specific bioactivity of therapeutics, such as biomaterials[21]. Others have utilized single-cell RNA sequencing (scRNAseq) to probe the immune[22-24] and fibroblast[22] responses due to biomaterial administration in volumetric muscle loss models, alongside using spatial transcriptomics to study biomaterials in wound healing[25,26]. However, we still lack an understanding of the regional and cell-specific responses to administering many types of therapeutics, including biomaterials, in MI.

Here, we combine spatial transcriptomics and snRNAseq to delineate pro-repairative bioactivity of an ECM biomaterial in subacute and chronic MI. Through local ECM administration, we determine regional pro-repair bioactivity within the ECM hydrogel. These cell-specific pro-repair programs are then measured per canonical cell types present in subacute and chronic MI. We measure immunomodulation, vasculature and lymphatic development, fibroblast activation, myocardial salvage and developmental gene activation, smooth muscle cell proliferation, and neurogenesis, all within canonical cell types within the heart. We then compare the pro-repair effects between subacute and chronic MI, where we find conserved repair bioactivity. Thus, we show how the combined transcriptomic technologies can provide insight into the pro-repairative effects of biomaterials to treat MI across time.

## Results

### ECM hydrogel elicits a spatially distinct pro-repairative effect in a subacute MI model

The ECM hydrogel was fabricated and characterized as previously described[9-11,27] [Supplementary Fig. 1A–F]. To measure the effects of ECM hydrogel administration during the subacute period post-MI, we injected either ECM hydrogel or saline into the infarct 7 days after ischemia-reperfusion surgery in rats. Hearts were harvested 7 days post-injection [Fig. 1a]. All replicate sections were first stained with hematoxylin & eosin (H&E) and quantified for infarct size [Supplementary Fig. 1G]. Then, all replicate sections were stained with anti-alpha-actinin to visualize and confirm spatial localization of fluorescently labeled ECM hydrogel within the infarct [Supplementary Fig. 2A]. We then performed capture- and sequencing-based spatial transcriptomics on the adjacent section of fluorescently labeled ECM hydrogel to measure gene expression changes in the local infarct microenvironment. Individual samples were integrated and unsupervised clustering of Visium capture spots was performed [Supplementary Fig. 2B] within the infarct, which was subsetted by low expression of pan-cardiomyocyte genes, *Myh6* and *Tnnt2*, and defined as the infarct zone[18] [Supplementary Fig. 2C, D]. Similarly, we defined high pan-cardiomyocyte gene expression regions as border zone or remote zone[18] [Supplementary Fig. 2C, D]. After quality control of the samples [Supplementary Fig. 3A] and integration of the individual ECM hydrogel samples [Table 1], we then aligned the fluorescent image [Fig. 1b, representative] with the spatial transcriptomic image [Fig. 1c]

to determine whether the ECM hydrogel elicited a unique transcriptomic signature. Here, we found 2 unique, transcriptionally distinct clusters within the infarct containing the ECM hydrogel [Fig. 1c, red] and one that lacked the hydrogel [Fig. 1c, cyan]. The ECM hydrogel zone elicited upregulation in genes that modulate the immune response (*Spp1, Fat1, Lgals1, Islr*), where infarct-only clusters were upregulated in primarily pro-inflammatory genes (*Ly6e, Cfd*) [Fig. 1d]. ECM remodeling and turnover (*Mmp2, Col1a1, Col12a1*), cardioprotective genes (*Ccn1*), cytoskeletal-mediated genes (*Tnc, Sdc1, Sdc2, Csrp2*), and angiogenic genes (*Col3a1, Col8a1, Col8a2, Serpine1*) were also found to be upregulated in regions containing the ECM hydrogel, suggesting the material promotes a pro-reparative response in the infarcted heart. Differentially expressed genes from ECM hydrogel clusters and infarct only clusters were analyzed by gene ontology (GO) enrichment in which areas containing ECM hydrogel indicated immune regulation, alongside development [Fig. 1e], while infarct-only areas elicit GO terms involved in lowered cell motility, and negative regulation of cell communication, coagulation, and locomotion [Fig. 1f]. All differentially expressed genes are displayed in Supplementary Data 1. We then integrated saline spatial samples [Table 1], where we also elucidated similar findings by measuring spatial differences between ECM hydrogel and saline samples [Supplementary Fig. 4A] showing pro-repair is unique to ECM hydrogel treatment. For global Visium comparisons, all differentially expressed genes are displayed in Supplementary Data 2. We then compared infarcted areas without ECM hydrogel directly to infarct areas in saline samples, where we found higher inflammatory genes (*Ifi27l2b, Isg15, Ly6e*) in the ECM hydrogel samples [Supplementary Fig. 4B]. However, in general, there are few differentially expressed genes distinguishing the two, which signifies that there is augmented pro-repair expression in areas where the ECM hydrogel is contained. For these comparisons, all differentially expressed genes are displayed in Supplementary Data 3. In addition to evaluating the regional effects of ECM hydrogel administration, we compared the remote zones of ECM hydrogel treated hearts to ones treated with saline [Supplementary Fig. 4C]. Here, the ECM hydrogel elicited markers for the anti-inflammatory response (*Angptl4, Gata3, Bcl11b, Il1r2*), while saline exhibited higher pro-inflammation markers (*Ccl2, Ccl3, Ccl7, Csf1, Ccl19, Il17ra*), and apoptotic processes (*Spn, Pdpn, Casp12, Inhba*). These comparisons are displayed in Supplementary Data 4.

### ECM hydrogel elicits unique pro-repairative effects at the single cell level

Given these findings of immunomodulation, vascular development, cardioprotection, and ECM remodeling resulting from hydrogel administration, we then wanted to measure the overall cellular responses present because of ECM hydrogel administration at a single-cell level. With the same therapeutic timeline and method of administration [Fig. 1a], we performed single-nucleus RNA-sequencing (snRNAseq) on the LV free wall of hearts treated with ECM hydrogel or saline [Table 1]. After quality control was completed [Supplementary Fig. 3B], the datasets were integrated and clustered, and we identified various cell types such as endothelial cells, fibroblasts, cardiomyocytes, immune cells (macrophages and T-cells), neuronal cells, smooth muscle cells, and lymphatic endothelial cells [Fig. 2a], all of which are represented in the infarcted heart with marker genes already depicted in the literature[18,20]; we also depicted the top features of each cluster [Supplementary Fig. 5A, Supplementary Data 5]. The relative proportions of each cell type respective to treatment were also illustrated in Fig. 2b. To investigate whether the infarcts of ECM hydrogel administered hearts were enriched with pro-repairative macrophages compared to saline treatment, macrophages were identified by *Ptprc* + / *Siglec1*+ expression and subclustered, with markers of each subcluster [Supplementary Fig. 6A] and relative distributions [Supplementary Fig. 7A]. ECM hydrogel treatment elicited a unique macrophage subset

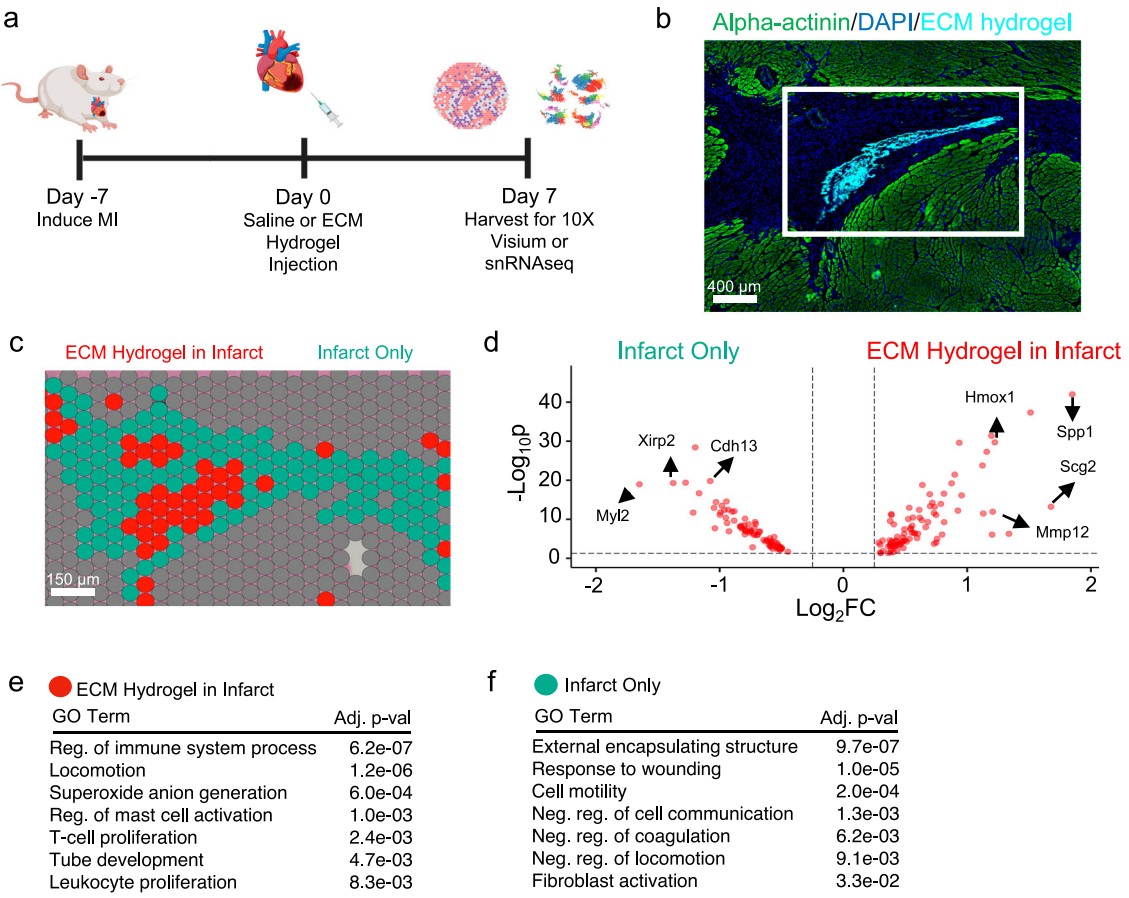

**Fig. 1 | ECM hydrogels demonstrate a spatially distinct transcriptomic profile, and induces an immunomodulatory and vascular development transcriptional protocol. a** MI was induced followed by an intramyocardial injection of ECM hydrogel or saline 7 days post-MI. Hearts were then harvested for either snRNAseq or spatial transcriptomics 7 days post-injection (14 days post-MI). Figure created in BioRender, and is licensed under CC BY 4.0 (https://biorender.com/r54nzgf). Sample size: $n = 2$ ECM hydrogel replicates, 7658 spots. **b** Myocardium (green) was labeled with anti-alpha-actinin antibody alongside fluorescently tagged ECM hydrogel (light blue) with nuclei stained with DAPI (blue). **c** The adjacent cryo-section was used for spatial transcriptomics via 10X Visium, where the infarct-containing ECM hydrogel (red) was found to cluster separately from the infarct alone (cyan). **d** The top upregulated differentially expressed genes defining the ECM hydrogel zone (red) were found to be immune and vascularly dominating genes compared to the downregulated genes impacting the infarct zone (cyan). **e, f** All differentially expressed genes in the ECM hydrogel zone (red) and infarct only zone were subjected to GO enrichment. Significance was determined via nonparametric Wilcoxon rank-sum tests with a Benjamini–Hochberg FDR adjustment to determine gene lists (**d**), and via Kolmogoro-Smirnov tests and permutation testing, with Benjamin-Hochberg FDR adjustment (**e, f**). Source data are provided as a Source Data file. ECM extracellular matrix, Neg negative, reg regulation, Pop population, Prolif proliferation, FC fold change.

**Table 1 | Overview of MI Models, Treatment Conditions, and Replicates for Different Transcriptomic Techniques**

|  | ECM hydrogel (subacute) | Saline (subacute) | ECM hydrogel (chronic) | Saline (chronic) |
|---|---|---|---|---|
| **snRNAseq Replicates** | 2 | 2 | 2 | 2 |
| **Total Nuclei** | 22230 | 18537 | 3160 | 2952 |
| **Spatial Replicates** | 2 | 3 | 3 | 2 |
| **Total Spots** | 7658 | 8036 | 9594 | 8166 |

[Fig. 2c, red] compared to macrophages from saline-treated hearts [Fig. 2c, cyan]. When we compared the unique macrophage subcluster from ECM hydrogel to that of the saline condition based off the upregulated genes with ECM treatment, we found a unique macrophage transcriptional signature (*Lyve1, Lgals3, Mrc1*), explaining the pro-reparative immunomodulatory response in the spatial data. Further, we found that the saline cluster was high in pro-inflammatory genes *(Cd300lb, Mertk, RT1-Db1, Mx1)* [Fig. 2c], alongside having GO terms enriched in inflammation and defense response [Supplementary Fig. 8A].

Importantly, we observed an interaction between the ECM hydrogel and T-cell activation [Fig. 1d] via *Fat1* and *Lgals1* expression.

In addition to a pro-reparative macrophage response, ECM hydrogels derived from other organs are known to promote a Th2 response through the mTOR/Rictor-dependence[28]. Given that the ECM hydrogel zone was enriched in T-cell proliferation, we subclustered T-cells (*Ptprc + /Itk +*), identified marker genes of these subpopulations [Supplementary Fig. 6B] with relative distributions [Supplementary Fig. 7B], and split them by treatment group [Supplementary Fig. 9A]. While there were no unique populations within each group, a global comparison between ECM and saline treated T-cells demonstrate that the ECM hydrogel specifically elicits a Th2 mediated response via upregulation of *Gata3*, a primary marker in Th2 activation[14], and by upregulation of *Il33*[29] and *Il1rl1*[30].

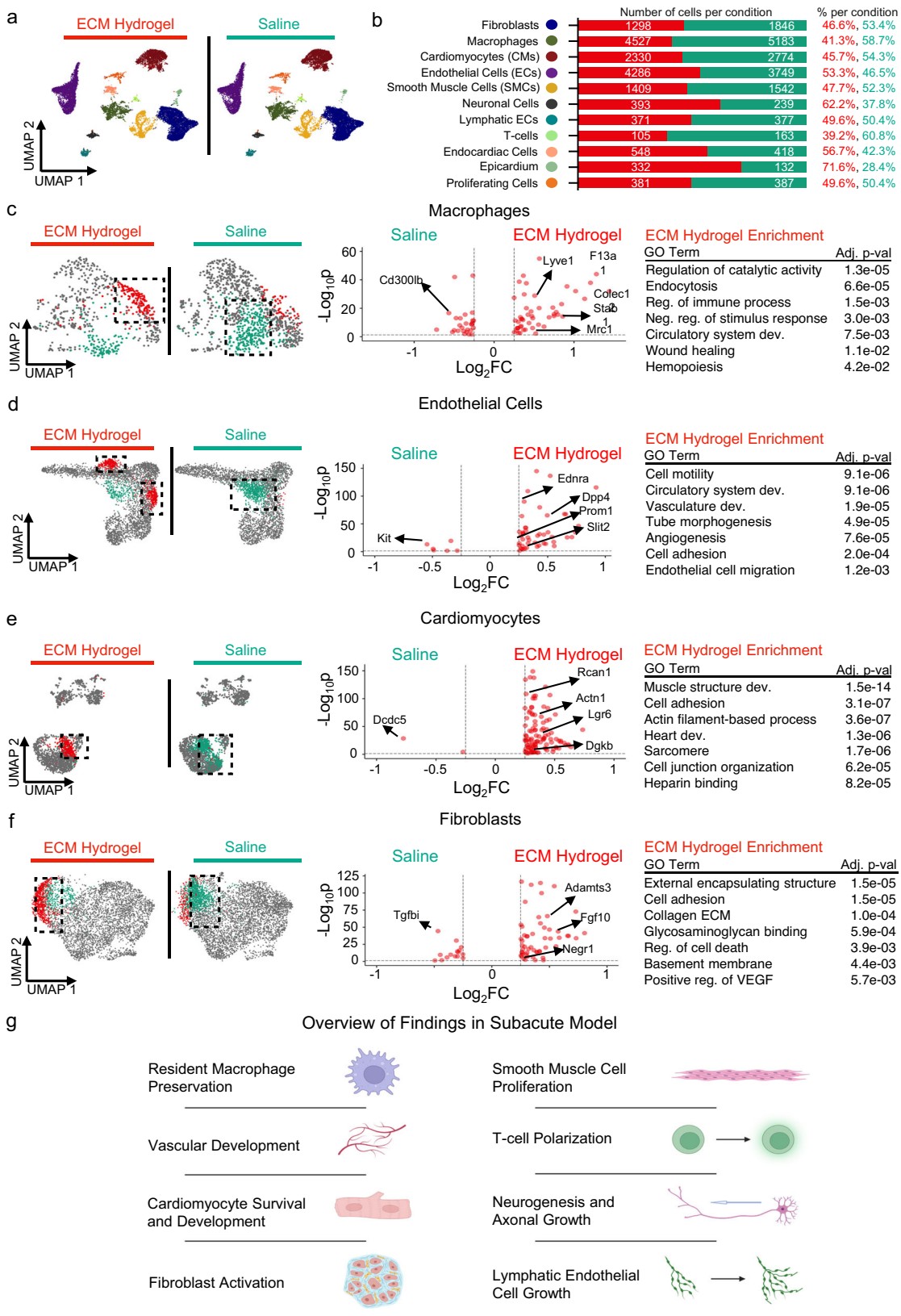

We have previously shown that the ECM hydrogel produces significant improvements in cardiac function[11,12,31], which may be linked to increased cardiomyocyte survival and proliferative potential[32], neovascularization[10,11], and reduction of collagen content[9]. Given that the spatial dataset reflected enrichment in tube development (*Col3a1*, *Col8a1*, *Col8a2*, *Serpine1*), we proceeded to investigate the endothelial cell specific changes present because of ECM hydrogel administration. After subclustering endothelial cells (*Cyyr1*+) and identifying marker genes of these subpopulations [Supplementary Fig. 6C] with relative populations [Supplementary Fig. 7C], we found 2 populations [Fig. 2d, red] specific to ECM hydrogel administration. Comparing these 2 clusters to a unique saline population [Fig. 2d, cyan], the upregulated

**Fig. 2 | ECM hydrogel promotes macrophage activation, neovascularization, cardiomyocyte development gene activation, and fibroblast activation in a subacute MI model. a, b** snRNAseq was performed on ECM hydrogel and saline hearts, where coarse clustering (**a**) defined primary cell types found in the heart with relative percentages and total number of cells per each primary cell time and per treatment (ECM hydrogel: red; saline: cyan) (**b**). Sample size: *n* = 2 replicates of subacute ECM hydrogel (22230 cells); *n* = 2 replicates of subacute saline (18537 cells). **c–e** Macrophages (**c**), endothelial cells (**d**), cardiomyocytes (**e**), and fibroblasts (**f**) were separately subsetted and reclustered into UMAP space. For each cell type, unique clusters to both the ECM hydrogel (red) and saline (cyan) groups were compared, with their differentially expressed genes displayed in a Volcano plot. All ECM hydrogel specific differentially expressed genes were subjected to GO enrichment. **g** Overview of findings with snRNAseq in the subacute model, where we found macrophage polarization, vascular development, cardiomyocyte survival and development, fibroblast activation, T-cell polarization, smooth muscle cell proliferation and development, neurogenesis, and lymphatic endothelial cell development. Figure created in BioRender, and is licensed under CC BY 4.0 (https://biorender.com/r54nzgf). Significance was determined via nonparametric Wilcoxon rank-sum tests with a Benjamini–Hochberg FDR adjustment to determine gene lists, and via Kolmogoro-Smirnov tests and permutation testing, with Benjamin-Hochberg FDR adjustment (**c–f**). Source data are provided as a Source Data file. ECM extracellular matrix, EC endothelial cell, CM cardiomyocyte, SMC smooth muscle cell, UMAP uniform manifold approximation and projection, reg regulation, Pop population, Prolif proliferation, Dev development, FC fold change.

genes found in ECM treatment were enriched in endothelial cell migration and blood vessel morphogenesis (*Adamts9, Vegfc, Sfrp2, Epha4*),.Within lymphatic endothelial cells, subsetted via *Prox1+/ Lyve1+* with subsetted marker genes [Supplementary Fig. 6D] and relative cell proprulations [Supplementary Fig. 7D], similar findings of increased vasculature development were found.

Comparisons of spatial datasets from the ECM hydrogel zone and the infarct-only zone in ECM treated animals revealed cytoskeletal-mediated genes, so we next focused on the cardiomyocytes within the snRNAseq dataset. After subsetting and reclustering the cardiomyocytes (*Rbm20+/Ryr2+*) from ECM hydrogel and saline treatment, we identified subclusters [Supplementary Fig. 6E], measured their relative proportions [Supplementary Fig. 7E], and discovered a hydrogel-associated cluster [Fig. 2E, red]. We then compared the unique ECM cluster with the saline-specific population (cyan) and found that the ECM hydrogel-specific cardiomyocyte subpopulation was distinguished by genes involved in cardioprotection and myocardial salvage (*Dgkb, Rcan1, Ndrg4, Ppara*), cardiac development (*Lgr6*), muscle structure development (*Des*), and cell junction organization (*Actn1*).

It has been also noted that decellularized ECM can modulate the fibroblast niche in MI[11] and in other disease phenotypes[22]. We also noted ECM remodeling genes within our corresponding spatial dataset [Fig. 1d]. After subclustering cardiac fibroblasts (*Dcn+/Gsn+*) and identifying subclusters [Supplementary Fig. 6F] with treatment proportions [Supplementary Fig. 7F], we found a hydrogel-associated fibroblast population [Fig. 2f, red] and we compared it to the unique saline population [Fig. 2f, cyan], where we found enrichment in formation of collagen containing extracellular matrix (*Fgf10, Colec12*), glycosaminoglycan binding (*Fgf10*), basement membrane organization (*Negr1*), and positive regulation of VEGF (*Adamts3*). GO analysis of the unique saline cluster highlighted oxidative phosphorylation [Supplementary Fig. 8B].

ECM hydrogels have been noted to promote endogenous brain tissue restoration by promoting cell infiltration in stroke models[33]. Given that the infarcted heart is known to attenuate neuronal responses, thus leading to arrhythmias[34], it is unknown how the ECM hydrogel modulates cardiac neuronal cells in the heart. We thus identified all neural cells (*Nrxn1+*) in the snRNAseq dataset, identified subcluster markers [Supplementary Fig. 6G] with treatment cell proportions [Supplementary Fig. 7G], and compared single-nucleus transcriptomes from ECM hydrogel [Supplementary Fig. 9C, red] and saline treatment [Supplementary Fig. 9C, cyan]. Genes upregulated in ECM hydrogel treatment were enriched in neurogenesis regulation (*Sema5a, Shtn1, Sema3c, Shank3*), cell projection organization (*Serpinf1, Reln, Sema3c*), neuron projection (*Magi2*), and insulin growth factor II binding (*Igfbp3, Igfbp4, Igfbp5*) [35].

Finally, given that smooth muscle cells are responsive to ECM cues[36], we also evaluated smooth muscle cell responses to ECM hydrogel administration, particularly in the infarcted heart. We thus subsetted all smooth muscle cells (*Tagln+/Acta2+*) from the snRNA-seq dataset and identified subcluster markers [Supplementary Fig. 6G]

with each treatment's proportions [Supplementary Fig. 7G], where we compared ECM hydrogel [Supplementary Fig. 9D, red] and saline treatment [Supplementary Fig. 9D, cyan]. Genes upregulated with ECM hydrogel treatment were found to be enriched in overall heart development (*Pdgfra, Angpt1*) and circulatory system development, alongside smooth muscle cell proliferation (*Ednra, Pdgfra*) and angiogenesis (*Agtr1a, Angplt1*). Saline-specific genes were found to be enriched primarily in metabolism-related pathways (*Cox4i1, Cox6a2*).

To verify our findings, we curated gene module scores from gene ontology (GO) of transcripts involved in the M1 and M2 responses, the Th1 and Th2 responses, angiogenesis, apoptosis, proliferation, and neurogenesis, particularly within our snRNAseq data. While there were no discernable differences in the M1 pro-inflammatory phenotype between ECM hydrogel and saline-treated hearts, there was an enriched M2 gene module score only present in ECM hydrogel-treated nuclei [Supplementary Fig. 10A]. While there was a nonsignificant increase in the Th1 response, we found that the ECM hydrogel-treated nuclei elicited a higher Th2 response than saline-treated nuclei [Supplementary Fig. 10B]. We also utilized the GO list for angiogenesis; we found increased vasculature development with ECM hydrogel administration in the snRNAseq dataset of subsetted endothelial cells [Supplementary Fig. 10C]. Within the cardiomyocyte dataset, we scored for apoptosis and proliferation, where we found lower cardiomyocyte apoptosis and higher cardiomyocyte proliferation within the ECM hydrogel treatment condition [Supplementary Fig. 10D and 10E]. Finally, among our neural cells, we found higher neurogenesis scoring with ECM hydrogel treatment [Supplementary Fig. 10F].

When comparing all our cells, we also ran ligand receptor analysis on the cell populations to determine uniform pathways exhibited by the treatment condition. In the ECM hydrogel condition, we found GO enrichment involving regulation of the Tgfβ2 pathway, T-cell activation, and the Igf signaling pathway [Supplementary Fig. 11A]. For saline cells, we found upregulation of the Nppa-Nrp1 ligand receptor complex, alongside GO enrichment in metabolic processes [Supplementary Fig. 11B]. Here, in the subacute model following ECM hydrogel administration, we were able to determine cell-specific responses, such as preservation of tissue-resident macrophages and macrophage polarization, increased vasculature development genes, promotion of a cardiomyocyte developmental program alongside decreased cardiomyocyte apoptosis, increased fibroblast activation, Th2 polarization, promotion of a neural cell developmental program alongside neural cell projection, and lymphatic development [Fig. 2g]. For each cell type, all marker genes of subclusters are displayed in Supplementary Data 6. All differentially expressed genes per cell type between ECM and saline are outlined in Supplementary Data 7.

### Response to ECM hydrogel is also spatially distinct in a chronic MI model

Previous studies of utilizing ECM hydrogels in chronic MI have yielded cardiac function improvement over time[12,13], alongside downstream gene expression effects, such as lowered macrophage response,

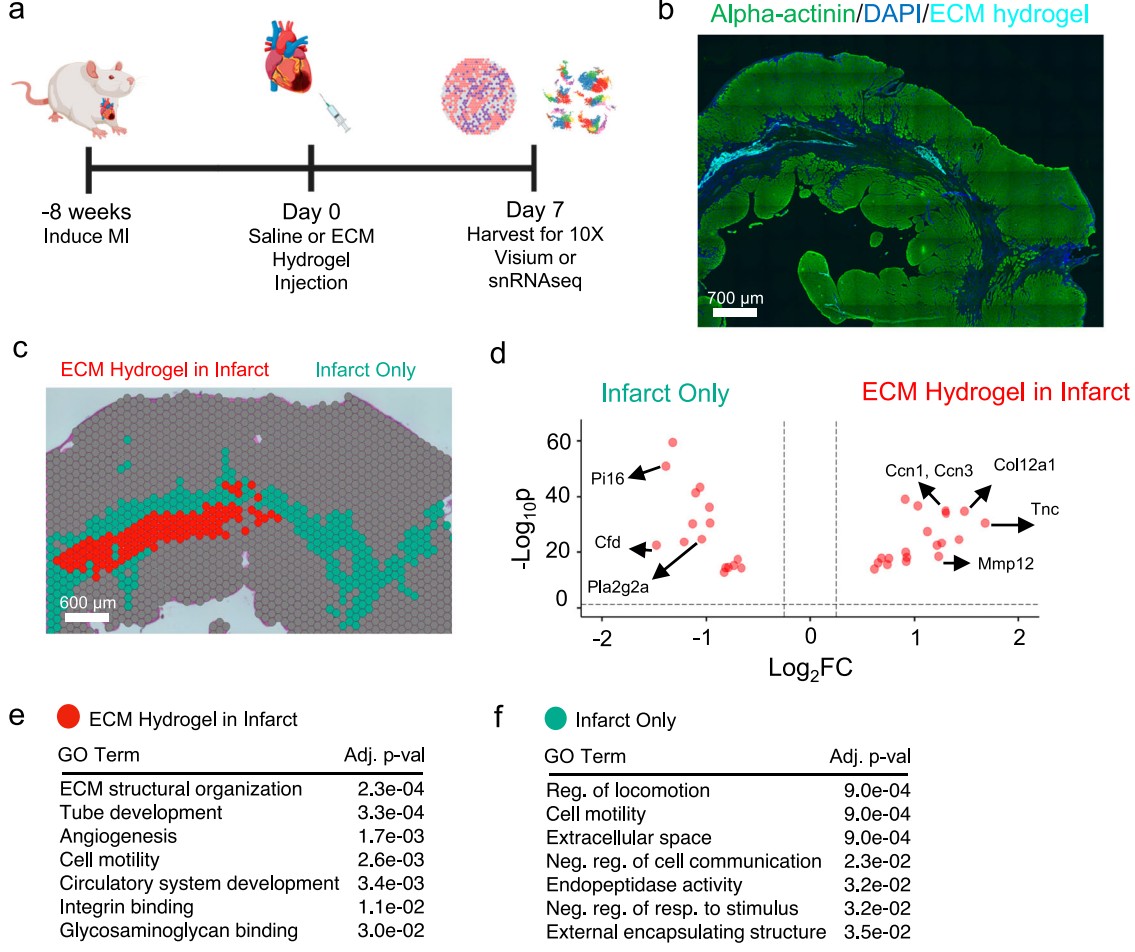

**Fig. 3 | ECM hydrogel administered in a chronic MI model also elicits spatial heterogeneity in the infarct. a** MI is induced followed by an intramyocardial injection of ECM hydrogel or saline 8 weeks post-MI. Hearts are then harvested for either snRNAseq or spatial transcriptomics 7 days post-injection. Figure created in BioRender, and is licensed under CC BY 4.0 (https://biorender.com/r54nzgf). Sample size: n = 3 ECM hydrogel replicates, 9594 spots. **b** Myocardium (green) was labeled with an anti-alpha-actinin antibody alongside fluorescently tagged ECM hydrogel (light blue). **c** An adjacent cryosection to the immunofluorescence image in (**b**) was used for spatial transcriptomics via 10X Visium, where the infarct containing ECM hydrogel (red) was found to cluster separately from the normal infarct zone (cyan). **d** Top differentially expressed genes for both ECM within infarct (red) and infarct alone (cyan) are shown. **e, f** A comparison of the two zones reflects the ECM hydrogel activates fibroblasts and is responsible for further vascular development, as demonstrated through GO enrichment. Significance was determined via nonparametric Wilcoxon rank-sum tests with a Benjamini–Hochberg FDR adjustment to determine gene lists (**d**), and via Kolmogoro–Smirnov tests and permutation testing, with Benjamin–Hochberg FDR adjustment (**e, f**). Source data are provided as a Source Data file. ECM extracellular matrix, Neg negative, reg regulation, Pop population, Prolif proliferation, FC fold change.

lowered Tgf-β signaling, and improvement of cardiac muscle contraction[12]. To further study the cell-specific and spatially distinct gene expression elicited by ECM hydrogel in chronic MI, we injected ECM hydrogel 8 weeks post-MI, and harvested 7 days post-injection [Fig. 3a, Table 1]. Replicate sections were first stained with hematoxylin & eosin (H&E) and quantified for sufficient infarct size[11] [Supplementary Fig. 1G]. Then, post quality control [Supplementary Fig. 3A], samples treated with fluorescently tagged ECM hydrogel [Supplementary Fig. 12A] were integrated, and unsupervised clustering of Visium capture spots was performed [Supplementary Fig. 12B] within the infarct, which was subsetted by *Myh6* and *Tnnt2* low expression as done earlier for subacute samples [Supplementary Fig. 12C/D]. We identified the fluorescently tagged ECM hydrogel area [Fig. 3b], as was done for the subacute model. With the adjacent cryosection, we then aligned both images. Using all ECM hydrogel spatial replicates, we then compared the ECM hydrogel spatial clusters [Fig. 3c, red] to the infarct-only clusters [Fig. 3c, cyan], with differentially expressed genes found for both regions and displayed via Volcano Plot [Fig. 3d]. Here, ECM hydrogel exhibits a strong fibroblast activation response via ECM reorganization (*Tnc, Actg2, Mmp12, Ccn2, Ccn3*), higher expression of

cell-junction genes (*Actb, Ccn3*), increased vasculature development (*Col8a1, Serpine1*), and increased cardioprotective/myocardial repair genes (*Col12a1*) [Fig. 3e]. Notably, pro-reparative genes from spatial analysis of subacute ECM administration (*Postn, Ccn2, Col8a1, Csrp2, Col12a1*) were also found in the chronic ECM spatial zone. The infarct only spatial cluster exhibited regulation of locomotion, negative regulation of cell communication, and endopeptidase activity [Fig. 3f], with all differentially expressed genes displayed in Supplementary Data 8. We also elucidated similar findings in the chronic model by measuring spatial differences between ECM hydrogel and saline samples [Supplementary Fig. 13A], with all differentially expressed genes displayed in Supplementary Data 9. Like in the subacute model, we compared infarcted areas without ECM hydrogel directly to infarct areas in saline samples, where we found the fibroblast remodeling gene (*Postn*) alongside the metabolic regulation gene (*Eef1a1*) in the ECM hydrogel samples [Supplementary Fig. 13B], further showing the pro-reparative response is localized to locations of the hydrogel, with all differentially expressed genes displayed in Supplementary Data 10. Finally, we compared the remote zones in ECM hydrogel-treated hearts to those in the saline group in the chronic model, where we

found genes involved in muscle development (*Cyp26b1, Col12a1, Col5a2, Tnn*) and cell cycle processes (*Cdc20, Ccne1, Ncaph, Smc2*) in the remote zone with ECM hydrogel treatment [Supplementary Fig. 13C]. In comparison, the saline remote zone exhibited significant inflammatory genes (*Ccl24, Ccl1, Tlr2*) and apoptotic processes (*Spn, Ptpn6, Il27ra, Pak1*), demonstrating that the ECM hydrogel also has significant effects in mitigating inflammation and apoptosis in the remote zone of the heart at chronic timepoints. For these comparisons, all differentially expressed genes are displayed in Supplementary Data 11.

### Administration of ECM hydrogel in chronic MI elicits cell specific pro-reparative genetic programs

We then performed snRNAseq on chronic MI samples following the same therapeutic timeline [Fig. 3a]. Cells were subjected to quality control [Supplementary Fig. 3B], clustered, classified with marker genes [Supplementary Fig. 5B, Supplementary Data 12], and split based off treatment condition [Fig. 4a]. Similar to the subacute model, we identified various cell types such as endothelial cells, fibroblasts, cardiomyocytes, macrophages, neuronal cells, smooth muscle cells, and lymphatic endothelial cells [Fig. 4a], with relative proportions of each cell type respective to treatment [Fig. 4b]. Similarly to the subacute model, we subsetted macrophages, identified subcluster markers [Supplementary Fig. 14A] and relative treatment populations [Supplementary Fig. 15A], and evaluated the differences between each treatment group [Fig. 4c]. We then compared the subsetted macrophages between ECM hydrogel [Fig. 4c, red] and saline [Fig. 4c, cyan], where we found higher macrophage polarization (*Spp1*) via enrichment in cytokine mediated signaling pathways (*Cx3cr1*) and regulation of monocyte chemotaxis (*Ccr1, Ccl2*).

Given that the spatial data elicited findings in vasculature development due to ECM hydrogel administration, we then looked at the corresponding endothelial cells between each treatment group [Fig. 4d]. After subclustering, identifying subcluster markers [Supplementary Fig. 14B] and splitting based off treatment groups [Supplementary Fig. 15B], we found a unique saline population [Fig. 4d, cyan] and directly compared to a population with higher ECM nuclei representation [Fig. 4d, red]. Among the genes that were upregulated with ECM hydrogel treatment, there was significant enrichment in actin-mediated cell contraction (*Actc1*), and response to oxygen containing compounds (*Epha4, Pdgfrb, Ncam1*). We also subsetted, reclustered out lymphatic endothelial cells, identified subcluster markers [Supplementary Fig. 14C], measured treatment specific cell proportion [Supplementary Fig. 15C], and separated them by condition [Supplementary Fig. 16A], where we found significant enrichment in tube development, angiogenesis, and vasculature development (*Pde1c, Cmahp, Slit2*), showing vasculature development within only lymphatic endothelial cells.

We next investigated the cardiomyocyte response due to ECM hydrogel administration [Fig. 4e]. After subsetting, reclustering, identifying marker genes in subclusters [Supplementary Fig. 14D] and measuring treatment-specific proportions in subclusters [Supplementary Fig. 15D], we found unique clusters in ECM hydrogel [Fig. 4e, red] versus saline [Fig. 4e, cyan] treatment, with the ECM hydrogel upregulated genes being with enriched in developmental growth (*Lgr6, Ncam1, Nrg1*) and neuron development (*Sema5a, Pak1, Nrxn1*). To thus verify this finding, we also subsetted, reclustered out neuronal cells from our snRNAseq chronic dataset, identified subcluster marker genes [Supplementary Fig. 14E], measured treatment-specific subcluster differences [Supplementary Fig. 15E], and compared between ECM hydrogel and saline [Supplementary Fig. 16B], where we found upregulation of genes involved in neuron projection, neuron development, and neurogenesis (*Nrxn3, Myh11, Chst11*). Given that neurogenesis is inhibited by MI[37], the presence of neurogenesis genes alongside cardiac development genes suggests that the ECM hydrogel

plays a role in promoting a neurogenic and developmental genetic program in the chronic MI model.

Spatial analysis of ECM hydrogel treatment also demonstrated ECM organization via expression of fibroblast-related genes. Thus, we probed into the fibroblast response present in the chronic MI heart, where we subsetted chronic fibroblasts from our single nucleus dataset, reclustered them, identified subcluster marker genes [Supplementary Fig. 14F], measured subcluster-specific proportions among treatments [Supplementary Fig. 15F], and compared between ECM and saline [Fig. 4f]. Compared to saline [Fig. 4f, cyan], ECM hydrogel-treated fibroblasts [Fig. 4e, red] expressed genes that were found to be enriched in supramolecular fiber organization (*Cst3, Actn2*), circulatory system development (*Ptn, Nox4*), and locomotion (*Igfbp3*).

In a similar fashion to the analyses conducted in the subacute model, we also evaluated the effect of ECM hydrogel administration on smooth muscle cells. Thus, we subsetted out smooth muscle cells from our snRNAseq chronic dataset, reclustered them, identified subcluster marker genes [Supplementary Fig. 14G], measured treatment-specific subcluster populations [Supplementary Fig. 15G], and compared between ECM and saline [Supplementary Fig. 16C]. Interestingly, we measured upregulation in developmental processes (*Trpc6, Itgb8*) alongside muscle hypertrophy (*Nfatc2, Cobl*). In comparison, saline had upregulation in vasculature development and angiogenesis (*Nrp2, Cav1*), signifying that ECM chronic administration may not benefit smooth muscle cells compared to saline.

Finally, we also performed ligand-receptor analyses for cells treated in the chronic phase of MI. With ECM hydrogel treatment, we found ligand receptor complexes involving *Vegfa*, which was corroborated with GO enrichment in angiogenesis [Supplementary Fig. 18A]. For saline cells, similar to the subacute model, we found upregulation of the ligand receptor complex Nppa-Npr1, alongside GO enrichment in metabolic processes [Supplementary Fig. 18B]. Taken together, ECM hydrogel administration in the chronic MI model thus demonstrated endothelial cell projection, promotion of a cardiomyocyte developmental program and neural cell projection, and fibroblast migration and ECM organization [Fig. 4g]. For each cell type, all marker genes of subclusters are displayed in Supplementary Data 13. All differentially expressed genes per cell type and direction are outlined in Supplementary Data 14.

### ECM Hydrogel Pro-Repair Effect Is Conserved Across Administration Timepoints

Next, we wanted to directly compare the pro-reparative effect of the ECM hydrogel across the two delivery time points. Thus, at the spatial level, we integrated the ECM hydrogel samples in the subacute and chronic MI models, and compared these ECM hydrogel zones to areas with infarct alone. The differentially expressed genes were then displayed on a Volcano Plot [Fig. 5a], where we found similar gene responses (*Tnc/Col12a1/Lox/Spp1/Csrp2/Serpine1*) that were found by looking at the subacute and chronic ECM administration models separately. The marker genes are displayed in Supplementary Data 15. GO analysis yielded similar findings to what was individually found via spatial analysis in the subacute and chronic models [Supplementary Fig. 19A]. We also compared the subacute and chronic zones of ECM, which found a higher response of differentially expressed genes, which were involved in development and response to wounding through GO analysis [Supplementary Fig. 19B]. These genes are displayed in Supplementary Data 16. With samples treated with ECM hydrogel, we also compared the subacute and chronic remote zones. Here, we found higher inflammatory processes (*Gata3, Lilrb4*) and development (*Vegfd, Pgf, Ccn3, Serpine1*) in the subacute model, with less differentially expressed genes in the chronic model [Supplementary Fig. 19C]. These findings are displayed in Supplementary Data 17. To further confirm our findings, after integrating the subacute and chronic ECM samples, we then displayed the average expression of the spatially

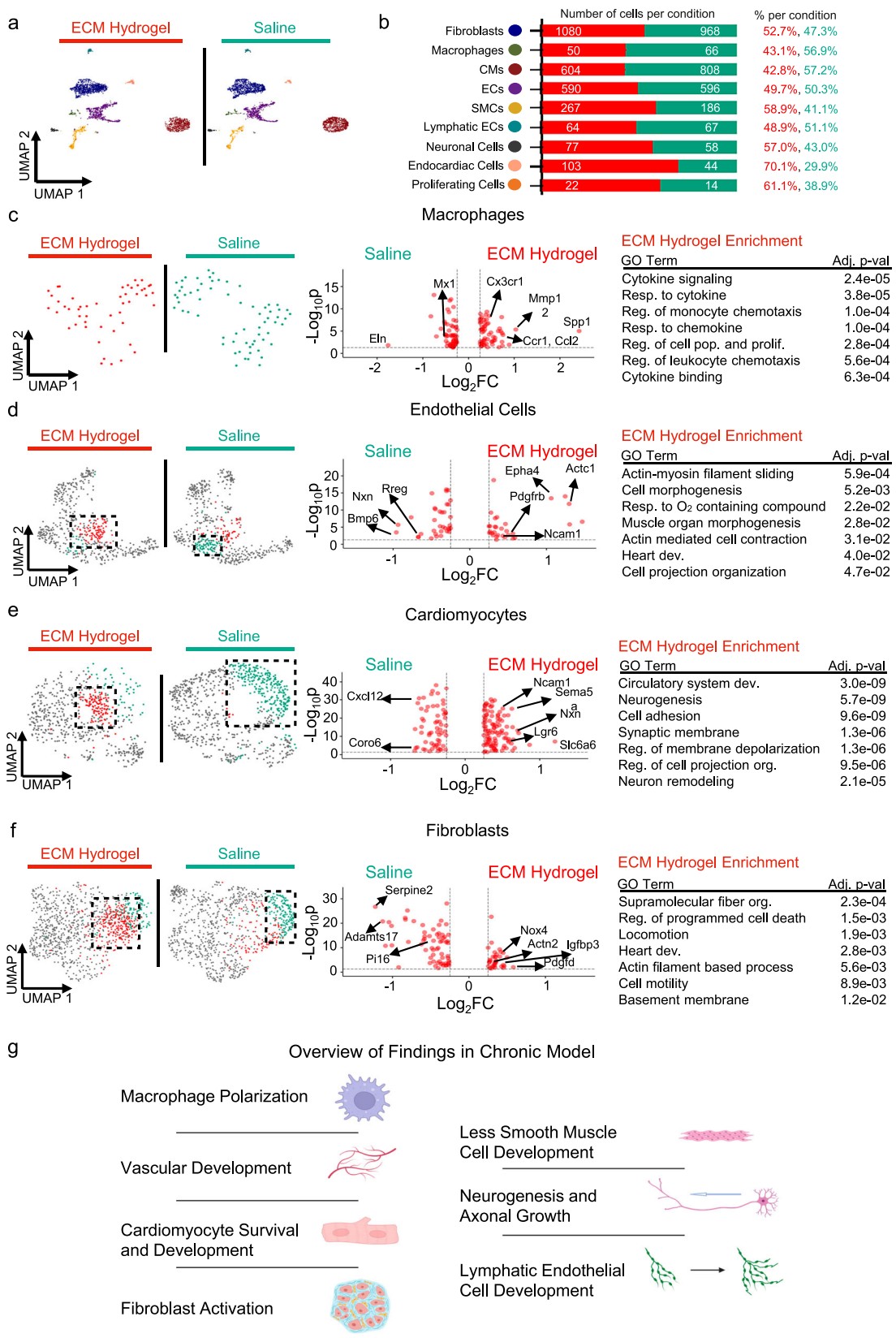

**g** Overview of Findings in Chronic Model

Macrophage Polarization

Vascular Development

Cardiomyocyte Survival and Development

Fibroblast Activation

Less Smooth Muscle Cell Development

Neurogenesis and Axonal Growth

Lymphatic Endothelial Cell Development

distinct matrix genes and mapped them to the previously defined subacute and chronic ECM zones, demonstrating higher expression in those zones [Fig. 5b]. Thus, we can infer marker biomaterial genes that are intrinsic to ECM hydrogel administration in MI.

To further decouple these findings found spatially, we wanted to measure overt cellular differences because of ECM hydrogel administration in the subacute and chronic MI models. Both models of MI were able to elicit downstream pro-reparative responses via immunomodulation, fibroblast activation, vascular development, and cardioprotection. Given that the cell numbers yielded from each model were variable, we downsampled the subacute and chronic models to ensure that the ECM hydrogel samples were comparable. To

**Fig. 4 | ECM hydrogels administered in a chronic model also promote vascular development, cardiomyocyte development gene activation, and fibroblast activation at the single cell level. a, b** snRNAseq of the chronic model was done to compare therapeutic effects between ECM hydrogel and saline, where coarse clustering (**a**) defined primary cell types found in the heart with relative percentages and total number of each primary cell separated by treatment group (**b**). Sample size: *n* = 2 replicates of chronic ECM hydrogel (3160 cells), *n* = 2 replicates of chronic saline (2952 cells).**c–f** Macrophages (**c**), endothelial cells (**d**), cardiomyocytes (**e**), and fibroblasts (**f**) were separately subsetted and reclustered into UMAP space. For each cell type, unique clusters to both the ECM hydrogel (red) and saline (cyan) were compared, with their differentially expressed genes displayed in a Volcano plot. All ECM hydrogel specific differentially expressed genes were subjected to GO enrichment. **g** ECM administration in the chronic model thus promotes macrophage polarization, vascular development, cardiomyocyte survival and development, fibroblast activation, smooth muscle cell proliferation and development, neurogenesis, and lymphatic endothelial cell development. Figure created in BioRender, and is licensed under CC BY 4.0 (https://biorender.com/r54nzgf). Significance was determined via nonparametric Wilcoxon rank-sum tests with a Benjamini–Hochberg FDR adjustment to determine gene lists, and via Kolmogorov–Smirnov tests and permutation testing, with Benjamini–Hochberg FDR adjustment (**c–f**). Source data are provided as a Source Data file. ECM extracellular matrix, EC endothelial cell, CM cardiomyocyte, SMC smooth muscle cell, UMAP uniform manifold approximation and projection, reg regulation, Resp response, Org organization, Pop population, Prolif proliferation, FC fold change.

evaluate conservation of immunomodulation, we subsetted macrophages through marker genes mentioned previously, and globally compared expression between both groups, where we found more homeostatic resolution in the subacute model vs. the chronic model, which had higher pathological immune responses [Fig. 5c]. As a result, this demonstrated that more immune resolution and modulation was present in the subacute model compared to the chronic model. However, subsetted and reclustered endothelial cells from subacute and chronic ECM hydrogel administration demonstrated conserved endothelial cell development and endothelial cell migration [Fig. 5D]. Uniquely, cardiomyocytes from the chronic timepoint elicited more cardiac development and neurogenesis markers. Subacute ECM treated cardiomyocytes did express higher *Acta1*, *Gpx3*, *Mfap5*, and *Uqcr10*, but elicited no GO enrichment [Fig. 5e]. Finally, within the fibroblasts, both subacute and chronic administration demonstrated overall ECM remodeling and fibroblast organization [Fig. 5f]. At the spatial level, we thus measured pro-repair effects that were conserved between subacute and chronic administration; however, we measured higher immunomodulation in the subacute model alongside higher cardiac development markers in the chronic model, demonstrating differences in timepoint administration. All differentially expressed genes across each cell type are included in Supplementary Data 18.

## Discussion

Previously, single cell and spatial transcriptomics approaches have been used in the heart to study overt pathological differences within MI[18,19], mechanisms of end stage heart failure[17], or generate general atlases of the heart[38], but have not been previously used to uncovering the regional and cell specific bioactivity of a therapeutic. In this study, the bioactivity behind injectable decellularized ECM biomaterials showed regioselective differences alongside specific cell type differences within the infarcted heart. While others have shown spatial transcriptomic effects due to biomaterial administration in non-cardiac indications[25,26], this study utilizes coupled spatial and single-nucleus analyses to further elucidate the effects of biomaterial therapeutics in MI.

Within these findings, we found that the ECM hydrogel elicits a spatially distinct genetic program in both subacute and chronic MI models through spatial transcriptomics. We were able to validate previous findings[10–12] that the ECM hydrogel was able to elicit immunomodulation within macrophages, vasculature development, promotion of a myocardial developmental program, and myocardial salvage, all of which are important aspects of a pro-repair response post-MI[39–41]. In addition to these findings, utilizing spatial transcriptomics showed a pro-reparative transcriptomic program that is spatially distinct in areas containing the ECM hydrogel in both MI models. Building upon other work in the field[28] that elucidated the remote effects of another ECM biomaterial treatment, we utilized spatial transcriptomics to further elucidate regional bioactivity in the remote zone of the heart. Thus, we were able to demonstrate anti-inflammatory effects in the remote region of ECM-treated hearts, alongside less pro-inflammatory and apoptotic effects. Moreover, snRNAseq showed preservation of tissue resident macrophages alongside macrophage polarization, T-cell polarization, axonogenesis and neurogenesis in neural cells, and lymphatic development, which can directly mitigate the pathological effects seen in MI[37,42–44]. In particular, it is noteworthy that the tissue resident macrophages (*Ptprc +* / *Lyve1 +* ) were found to have higher expression in ECM treatment alongside exhibiting anti-inflammatory markers (*Cd163 +* ) as typically in MI, recruited macrophages exhibit a more pro-inflammatory state compared to the tissue-resident macrophages, which exhibit a more anti-inflammatory phenotype[42,45]. In addition, we measured a Vegf-secreting fibroblast population, which was never detected in our previous analyses, but other studies have shown cardiac fibroblasts can secrete Vegf to elicit vascular development[46,47].

While others have measured T-cell polarization due to biomaterial administration in other tissues[28], no one has measured T-cell polarization in the heart post-ECM biomaterial administration. The anti-inflammatory T-cell response, indicated by *Gata3*+ in Cd4 T-cells, is known to induce pro-repair with ECM biomaterials in other disease models[28]. In addition to the anti-inflammatory macrophage response present in the subacute model, we were able to measure a significant anti-inflammatory helper T-cell response, demonstrating multiple paths for immunomodulation, an important facet of the pro-repair response post-MI[39]. In addition, to our knowledge, no one has studied decellularized ECM biomaterial effects on lymphatics in general, and in neural cells within the heart. We note here that a significantly upregulated gene with ECM treatment in our lymphatic endothelial cells was *Sox13*, which is known to mediate the inflammatory effects present in the endothelium[48]. Among neural cells, which were subsetted via *Nrxn1 +* , ECM hydrogel treatment had high semaphorin (*Sema5a*) signaling and relin (*Reln*); semaphorin is known to play a significant role in cardiovascular development[49] and axon guidance[50], while relin is known to improve cardiac function and repair by improving neuronal migration[51].

In both the subacute and chronic MI models, we thus see conserved therapeutic bioactivity and pro-reparative programs at the spatial level. While we acknowledge that evaluation at more acute time points may gather further insight into the ECM hydrogel's mechanism of action, we note that this timepoint was previously shown to have peak cell infiltration into the ECM hydrogel, which would be the best timepoint to measure cellular and transcriptional changes[9], and previous ECM hydrogel studies measuring gene expression changes peaked at 7 days post-injection[11]. Thus, we compared the findings between the subacute and chronic timepoints at 7 days post-injection to see if ECM pro-repair response was conserved across timepoint administration. When integrating the subacute and chronic model ECM zones, there were common collagen related genes (*Col8a1*, *Col12a1*), ECM deposition genes (*Ccn2*, *Thbs4*, *Tnc*), wound healing genes (*Spp1*, *Lox*, *Enpp1*), and development genes (*Col8a1*, *Col8a2*, *Csrp2*), demonstrating that between both models, there are unique

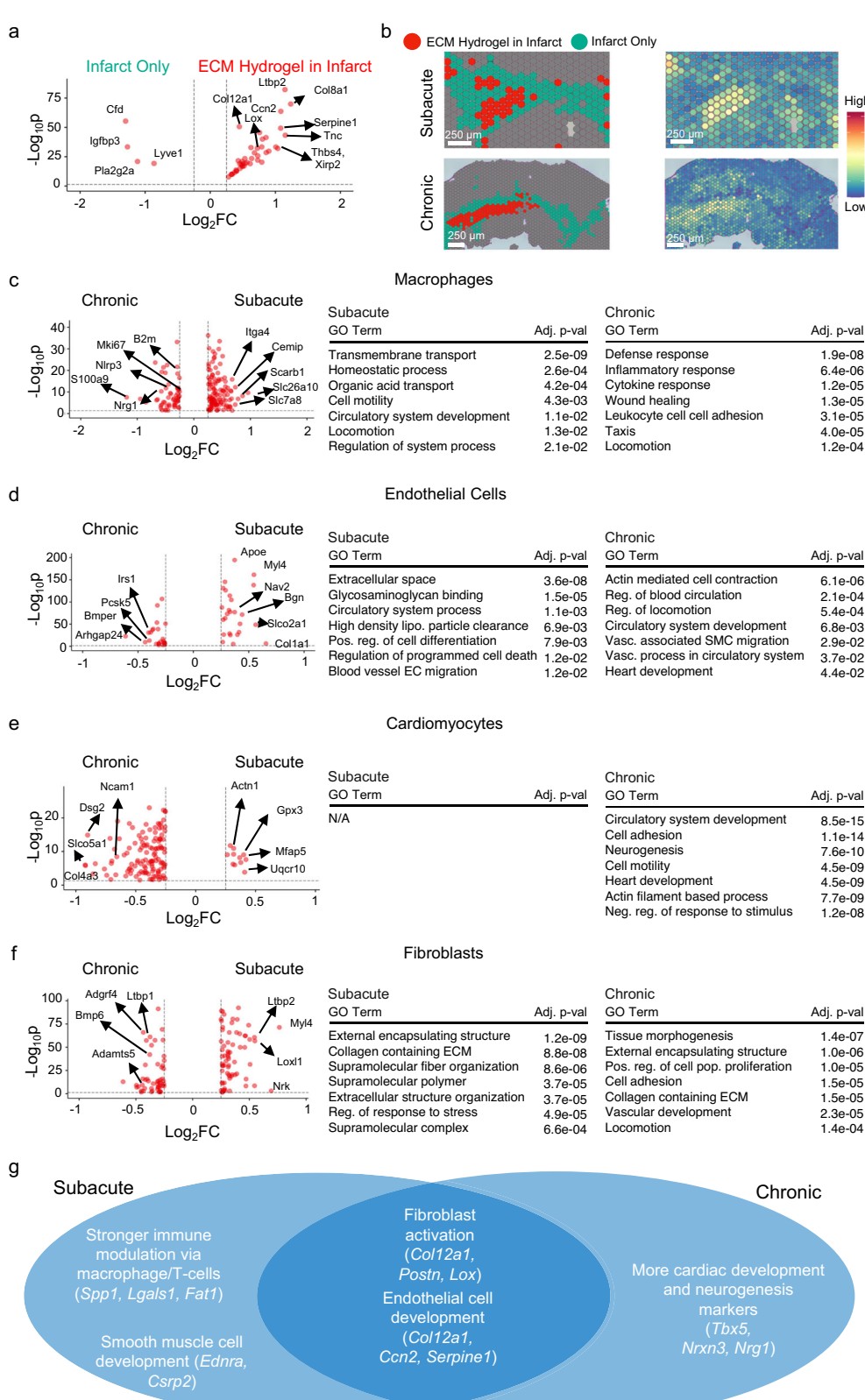

marker genes that are unique to ECM hydrogel administration, and directly demonstrate how the ECM hydrogel elicits cues on its surrounding microenvironment in the infarcted myocardium post-MI. Through spatial transcriptomics, we also demonstrated how the ECM hydrogel can elicit cues in the remote myocardium when comparing between the subacute and chronic timepoints, demonstrating that the ECM hydrogel can elicit pro-repair responses in remote areas of the

heart. Finally, we also note that certain genes *(Csrp2/Col12a1)* have continuously appeared as differentially expressed genes in both MI models presented in this manuscript and heart failure models[52] with ECM hydrogel treatment. *Col12a1* has recently been shown to elicit cellular development alongside tissue repair and regeneration[53,54]. Here, it is standard for both snRNAseq and spatial transcriptomics to gain effective power when they are cross-validated, especially when

**Fig. 5 | ECM hydrogel administration elicits similar transcriptomic programs regardless of timepoint administration but different cellular responses. a** The top upregulated differentially expressed genes defining the ECM hydrogel zone (red) with integrated subacute and chronic Visium were found to be immune, fibroblast, and vascularly dominating genes compared to the downregulated genes impacting the infarct zone (cyan). Sample size: *n* = 2 for subacute ECM hydrogel (7658 spots); *n* = 3 for chronic ECM hydrogel (9594 spots) **b** The ECM zones in both subacute and chronic models of MI have higher expression of the matrix specific genes relative to the infarct zone. **c**–**f** Macrophages (**c**), endothelial cells (**d**), cardiomyocytes (**e**), and fibroblasts (**f**) treated with ECM hydrogel in subacute and chronic MI were subsetted, reclustered, and compared with respect to MI

timepoint. Sample size: *n* = 2 subacute ECM hydrogel (downsampled to 3000 cells), *n* = 2 chronic ECM hydrogel (downsampled to 3000 cells). Top differentially expressed genes were displayed via Volcano Plot, and the differentially expressed genes were subjected to GO enrichment. **g** Comparison of transcriptomic findings between subacute and chronic MI. Significance was determined via nonparametric Wilcoxon rank-sum tests with a Benjamini−Hochberg FDR adjustment to determine gene lists (**a**, **c**–**f**), and via Kolmogorov−Smirnov tests and permutation testing, with Benjamini−Hochberg FDR adjustment (**c**–**f**). Source data are provided as a Source Data file. ECM extracellular matrix, neg negative, vasc vascular, pos positive, reg regulation, pop population, prolif proliferation.

administering the same therapeutic in different models of MI. Thus, we can demonstrate significant therapeutic pathways that are conserved with ECM hydrogel administration in MI through spatial transcriptomic and snRNAseq analyses.

We also noted conserved pro-repair responses within fibroblasts and endothelial cells, but differences were observed with immunomodulation, smooth muscle cell development, and cardiac development responses between MI models. While immunomodulation was able to be detected when directly comparing the two models, we noted that the subacute model elicited overall resolution (*Spp1*) compared to the pro-inflammatory environment in chronic MI administration (*S100a9, Nlrp3, Ifitbl1*). We also noted that smooth muscle cell proliferation and development (*Ednra, Agtr1a*) were only detected in the subacute model; uniquely, the chronic model exhibited more oxidative stress markers with ECM hydrogel administration. Thus, markers of pro-inflammation and oxidative stress present at later timepoints treated with ECM may also indicate a diminished pro-repair response at the chronic timepoint[55]. However, it is important to note that the cardiomyocytes in the chronic model were able to elicit more cell adhesion (*Dsg2, Ncam1*), development (*Tbx5*), and calcium transporter (*Slc6a6, Slco5a1*) responses, with neuronal cells exhibiting neurogenic markers (*Nrg1, Nrxn3, Slc9a9*). Thus, while the chronic timepoint may exhibit more markers of oxidative stress and pro-inflammation, chronic ECM hydrogel administration increased developmental and neurogenic markers, signifying other ways that the ECM biomaterials can still induce pro-repair through neural[56] and cardiac[57] development.

Taken together, these results suggest that ECM biomaterials modulate multiple cell types in the infarcted heart and allow for an overall pro-reparative response in treating MI. Thus, by modulating the immune response, promoting Vegf-producing fibroblasts, increasing vasculature development, eliciting cardiomyocyte development and salvage, and promoting smooth muscle cell proliferation, ECM biomaterials can thus lead to a pro-repair response. In addition, we are the first to study the effects of administering ECM hydrogels on other cell types. While we note that GO analyses should be interpreted with caution, the GO term results are consistent with previously published findings, including neovascularization[8,11], immunomodulatory processes[14,58], fibroblast-mediated responses[59], myocardial salvage and developmental gene activation[8,10,11], and smooth muscle cell proliferation[59]. We also found that ECM hydrogels promote lymphatic endothelial cell development and neurogenesis. While a limitation of this study is the lack of other analyses on these cell types, these transcriptomic studies prompt future studies on how ECM biomaterials impact lymphangiogenesis and neurogenesis in tissue. Finally, while we note that the ECM hydrogel exhibits individual effects per cell type, we were also able to measure a global pro-repair phenotype to narrow down how the ECM hydrogel affects biological pathways. Through ligand-receptor analysis and GO terms of the ligand-receptor pairings, we thus determined that the ECM hydrogel reduces the inflammatory response and reduces Tgfβ signaling in the subacute model, alongside increasing angiogenic potential in the chronic model, demonstrating that a heterogeneous mixture of ECM proteins can elicit significant

effects on mediating pathways upregulated in MI. Finally, we found the ECM hydrogel downregulating a particular pathway involving *Nppa* and *Npr1*, markers that are involved in MI autophagy, particularly in cardiomyocytes[60,61], which is a potential avenue for future studies.

Utilizing tools such as snRNAseq and spatial transcriptomics allows one to investigate the regional and cell-specific bioactivity of therapeutics such as injectable ECM biomaterials. While these transcriptomic tools can still be further improved upon, the current resolution was able to confirm and further show how injectable biomaterials and generally therapeutics elicit a pro-reparative effect in MI. While our current study lacks assessment of cardiac function, we have already demonstrated benefits in cardiac function with the ECM hydrogel in both subacute[10,11] and chronic[12] rat MI models via MRI and a subacute pig MI model via echocardiography. [9] Here, we utilized infarct size via histology as our metric to exclude animals that did not have a sufficient infarct, which we routinely do even in studies that use cardiac imaging[10–12]. While our sample sizes are comparable to others in the field[22,25], we acknowledge that a limitation of our study is a modest sample size due to the costly nature of higher end transcriptomic techniques. However, by coupling these two methods together, we were able to demonstrate pro-reparative bioactivity of an ECM-based biomaterial administered at different timepoints post-MI through spatial transcriptomics and snRNAseq. Across the different models and materials, we found responses of immunomodulation, vasculature and lymphatic development, fibroblast activation, myocardial salvage and developmental gene activation, smooth muscle cell proliferation, and neurogenesis. Thus, we demonstrate the ECM hydrogel's pro-reparative effects in mitigating the MI response and promoting cardiac repair, and show the potential of using combined spatial and single nucleus transcriptomics to elucidate therapeutic pathways for MI treatments.

## Methods
### ECM hydrogel preparation and characterization
ECM hydrogel was prepared from porcine LV myocardial tissue[8,9,11]. Briefly, the tissue was chopped, decellularized in sodium dodecyl sulfate detergent, lyophilized, milled into a fine powder, subjected to partial pepsin enzymatic digestion in HCl, had its pH and salts adjusted, brought to a final concentration of 6 mg/mL, and finally aliquoted and lyophilized for storage at −80 °C. ECM hydrogel was resuspended at 6 mg/mL with sterile water with or without VivoTag™ 750 for fluorescent labeling[31,62]. ECM hydrogel was labeled at a final dilution of 1:3000.

All ECM hydrogel characterization was performed as below[9,11,27]. Double-stranded DNA (dsDNA) was obtained by performing the Picogreen (Life Technologies, Carlsbad, CA, USA) assay (*n* = 2). Sulfated glycosaminoglycan (sGAG) concentrations were quantified via the 1,9-dimethylmethylene blue (DMMB) (Sigma-Aldrich) assay. SDS-polyacrylamide gel electrophoresis (SDS-PAGE) was used to assess the protein and peptide content of the ECM hydrogel, with collagen as the control. The residual SDS from decellularization was detected using the *Residual SDS detection kit (Bio Basic)*. Rheometry was used to obtain the complex viscosity. Rheometry data was measured on an

ARES G2 Rheometer (TA instruments). Viscosity was measured at 25 °C using a 20mm flat geometry with 200 µL of sample and a gap height of 500 µm, 10 points per decade from 0.1 to 1000rad/s using a flow procedure with an equilibration time of 30seconds. For storage and loss moduli measurement, 500 µL of sample was loaded into a 4 mL scintillation vial and allowed to undergo gelation for 24 hours at 37 °C. After 24 hours, the storage and loss moduli were measured at 37 °C, from 0.1 to 100 Hz with 5 points per decade and a gap height of 1200 µm.

### Surgical Procedures

All procedures in this study were approved by the Committee on Animal Research at the University of California, San Diego, and in accordance with the guidelines of the Association for the Assessment and Accreditation of Laboratory Animal Care. All surgeries were performed using aseptic conditions. All animals used in the study were adult female Sprague Dawley rats (225 to 250 g). To induce MI, all animals underwent ischemia–reperfusion surgery to occlude the left main artery for 35 minutes[8,31]. At either 1 week post-MI (subacute) or 8 weeks post-MI (chronic), animals were subjected to a 75 µL injection of ECM hydrogel or saline with a 27 G needle directly into the infarct via subxiphoid access[10,12]. A single injection into the infarct is performed to minimize needle-based tissue damage and to directly treat the infarct.

### Tissue processing

All animals in the subacute and chronic models were harvested at 1 week post-injection (2 weeks and 9 weeks post-MI, respectively). Half of the animal hearts from each model group were cut into 6 slices using a stainless-steel rat heart slicer matrix (Zivic Instruments) with 1.0mm coronal spacing. Odd slices were frozen in TissueTek OCT™ and sectioned into 10 µm thick slices and placed onto a 10X Visium Spatial Transcriptomics Slide or a regular histology slide. Tissue sections adjacent to the Visium section were taken within 100 µm for IHC. For the other half of hearts, the LV free wall was isolated into even slices of tissue and flash frozen in liquid nitrogen to preserve RNA. The remaining slices were fresh frozen in TissueTek OCT freezing medium for histology.

### Histology and immunohistochemistry

Prior to running the 10X Visium Spatial Transcriptomics protocol, slides were stained with H&E and scanned at 20X using either a Nikon Ti2 microscope or an Olympus VS120 Slide Scanner. Immunohistochemistry was performed using an a-actinin antibody (aACT, Sigma 233 A7811, 1:700), with ECM hydrogel fluorescently visualized with VivoTag™ 750 and nuclei with Hoechst 33342 (Life 228 Technologies). Antibody validation was performed with primary only, secondary only, and isotype controls per bulk stain.

To visualize infarct size and tissue morphology, slides at evenly spaced locations spanning the ventricles were stained with hematoxylin & eosin (H&E), mounted with Permount (Fisher Chemical), and scanned at 20X using either Aperio Scan Scope CS2 slide scanner (Leica Biosystems) or an Olympus VS120 Slide Scanner. Infarcts were manually traced in QuPath to quantify infarct size by measuring the infarct size within the left ventricular myocardium compared to the total left ventricle[11,12].

### Nuclei Isolation and snRNAseq

Flash frozen LV free wall samples were resuspended in nuclei lysis buffer (Millipore Sigma, Nuclei EZ prep, NUC101), 0.2 U/µL RNase inhibitor (Enzymatic Y9240L), and finely chopped with scissors[18]. Samples were homogenized with a 2mL dounce grinder (KIMBLE) with the lysates filtered through 100 µm, 50 µm, and 20 µm strainers (CellTrics filters), respectively. The samples were centrifuged at 1000 g for 5 min at 4 °C to pellet nuclei. The nuclear pellet was resuspended in

a sucrose gradient buffer and centrifuged at $1000 \times g$ at 4 °C for 10 min. The pellet was then washed with a nuclei storage buffer containing 10 µg ml−1 4′,6-diamidino-2-phenylindole DAPI and centrifuged at 500 g and 4 °C for 5 minutes, with the pellet resuspended in 200 mL of 2% BSA in 1x PBS. Nuclei were then counted on a hemocytometer and volume adjusted to 1000 nuclei/µL for loading. snRNAseq was performed with a microfluidic droplet-based technique provided by 10X Genomics kits (Chromium Next GEM Single Cell 3′ GEM V3.1). Quality control for cDNA and libraries was performed on Aligent TapeStation, and library concentrations were determined via Qubit HS DNA kit. Paired-end sequencing was run on a NovaSeq6000 instrument. Demultiplexing of sequenced samples were mapped to a rat reference genome (Rnor6.0, including introns). Redundant unique molecular identifiers (UMIs) were eliminated via the Cell Ranger 7.1.0 pipeline from 10X Genomics. All snRNAseq samples are outlined in Table 1.

### Spatial transcriptomics

As mentioned before, harvested OCT-embedded cardiac tissue was cryosectioned onto the fiduciary regions on a 10X Visium slide at 10 µm thickness. Sections were then stained with hematoxylin and eosin, with images obtained on a Nikon Eclipse Ti2-E widefield microscope at 10X magnification. Visium was performed as per the manufacturer's kit instructions (Visium Spatial Gene Expression), with tissue permeabilization time of infarcted rat hearts optimized at 42 minutes. Both protocols utilized barcoding and library preparation, which was validated using an Agilent TapeStation prior to sequencing and quantified via Qubit HS DNA Kit. Paired-end sequencing was done on a NovaSeq6000 instrument. Low-level analysis was performed by mapping to a rat reference genome (Rnor6.0, including introns). The SpaceRanger 1.3 pipeline (10X Genomics) was used to remove redundant UMIs. All Visium samples are outlined in Table 1.

### Quality control, normalization, and integration

All snRNAseq and Visium data analysis was done using the Seurat package (v4) in R. Both sets of data had raw counts for each gene normalized to specific transcript count and log-transformed. Ribosomal and mitochondrial genes were filtered out as done previously[18], and cells/pixels with greater than 200 counts were retained for further analysis. Doublets were removed by determining cells that contained non-endogenous gene markers[18]. Highly variable genes across individual samples were determined via the FindVariableFeatures method from Seurat R package v4 to find the top 2000 genes with the highest feature variance.

For analyses between ECM hydrogel and saline treatment groups in snRNAseq, integration of all snRNAseq replicates was performed in Seurat to enable harmonized clustering and downstream comparative analyses[18]. Canonical correlation analysis (CCA) was used to determine anchoring cell pairs, and integration anchors were detected through the FindIntegrationAnchors function, utilizing reference-based integration of the single-nucleus datasets via CCA. Once the ECM and saline datasets were integrated together, the integrated set was then subjected to principal component analysis, and further reduced through uniform manifold approximation and projection (UMAP) to subject the data into a consensus UMAP space. The data was then clustered and visualized in UMAP space. All canonical cell types (macrophages, endothelial cells, cardiomyocytes, fibroblasts, T-cells, neuronal cells, smooth muscle cells, and lymphatic endothelial cells) were identified through literature review of the gene signatures[38], alongside using the FindAllMarkers function in Seurat v4 ($P_{adj}$ <0.05, logFC > 0.25, min.diff.pct > 0.25, assay = RNA) [Supplementary Fig. 4, Supplementary Data 5, Supplementary Data 12]. Spatial transcriptomic analyses of ECM and saline treatment groups were subjected to the same integration process to ensure comparisons were done in a consensus UMAP space.

For each relevant cell type, the data were subsetted through gene markers, integrated via CCA, renormalized, subjected to principal component analysis, and reduced through UMAP to subject each cell type across treatment conditions into a consensus UMAP space. Integration for each cell type in the subacute model is depicted in Supplementary Fig. 6. Integration for each cell type in the chronic model is depicted in Supplementary Fig. 14.

### Coarse clustering and differential gene expression analyses

snRNAseq data were integrated as described above and clustered (resolution = 1). As mentioned above, each canonical cell type (macrophages, endothelial cells, cardiomyocytes, fibroblasts, T-cells, neuronal cells, smooth muscle cells, and lymphatic endothelial cells) was subsetted regardless of condition into a consensus 2D UMAP space. These subsets per cell type were subjected to further coarse clustering (resolution = 1). Subset-specific features per cluster lists were determined using FindAllMarkers ($P_{adj}$ <0.05, logFC > 0.25, min.diff.pct > 0.25), displayed as a heatmap with the top 5 features per cluster, and are displayed in Supplementary Data 6 for the subacute model, and Supplementary Data 13 for the chronic model. Visium data were integrated as described above and clustered (resolution = 1).

All analyses between ECM and saline were conducted using the FindMarkers function in Seurat v4 ($P_{adj}$ <0.05, logFC > 0.25, min.diff.pct > 0.25, assay = RNA [snRNAseq] or assay = Spatial [Visium]).

### Gene ontology enrichment

Gene ontology (GO) was performed using the gene set enrichment analysis software (UC San Diego and Broad Institute) which determined enriched pathways via nonparametric Kolmogoro–Smirnov tests and permutation testing, with Benjamin-Hochberg FDR adjustment[63,64]. Inputs for GO analyses were determined using the FindMarkers function in the R package Seurat ($P_{adj}$ <0.05, logFC > 0.25, min.diff.pct > 0.25, assay = RNA [snRNAseq] or assay = Spatial [Visium]) to find differential genes between ECM and saline cells.

### Gene expression scoring and ligand receptor analyses

Gene expression scoring was performed by summing the expression of genes specified under a specific pathway of interest from GO lists, was then scored by the summed expression of genes in a particular pathway (M1, M2, Th1, Th2, Apoptotic Processes, Cellular Proliferation, Neurogenesis, and Angiogenesis) and finally normalized to the total RNA content. All data were integrated and processed in R (version 4.3) with the Seurat package. sciDiffComm (version 1.1.1) was used to perform ligand-receptor analysis between conditions via overrepresentation analysis.

### Statistical analysis

Group size was based on previous literature of biomaterial single cell[22,23] and spatial studies[25]. Comparisons between two groups were analyzed using two-tailed Mann-Whitney nonparametric tests. Differential gene expression analyses were performed using nonparametric Wilcoxon rank-sum tests with a Benjamini–Hochberg FDR adjustment to determine gene lists all in Supplementary Data 1-18. Gene ontology was calculated using the gene set enrichment analysis software via Kolmogorov–Smirnov tests and permutation testing, with Benjamin–Hochberg FDR adjustment[63]. The numerical data were presented as violin plots, box plots, or box and whisker plots. Statistical significance was determined via $p < 0.05$, with (*$p < 0.05$, **$p < 0.01$, ***$p < 0.001$, ****$p < 0.0001$).

### Reporting summary

Further information on research design is available in the Nature Portfolio Reporting Summary linked to this article.

### Data availability

The single nucleus RNA sequencing and spatial transcriptomic sequencing datasets generated in this study have been deposited to the Gene Expression Omnibus data under accession GSE262430. All other data supporting the findings in this study are included in the main article and Supplementary file as Source Data. Source data are provided with this paper.

### Code availability

All R scripts used to analyze RNA-seq and sequencing-based spatial transcriptomic data are publicly available at Zenodo[66] (https://zenodo.org/records/16918242).

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

## Acknowledgements

This research was funded in part by the National Institutes of Health (NIH) National Heart, Lung, and Blood Institute (NHLBI) grants R01HL113468 (K.L.C.), R01HL146147 (K.L.C.), R01HL165232 (K.L.C.), and R01HL181539 (K.R.K.). This research was funded in part by an NIH National Institute of Arthritis and Musculoskeletal and Skin Diseases (NIAMS) grant DP2AR075321 (K.R.K.). This work was funded in part by an NHLBI Training Grant T32HL105373 (J.M.M., M.D.D., J.D.H., R.M.W., D.M.C.), an NHLBI Training grant T32HL007444 (V.K.N.), an NIBIB Training Grant T32EB009380 (A.C.), an NIH T32 Training Grant T32HL007444 (M.B.N), American Heart Association pre-doctoral fellowships 23PRE1023221 (J.M.M.) and 24PRE1180449 (A.C.), an AHA post-doctoral fellowship 24POST1242447 (M.B.N.), NHLBI pre-doctoral fellowships F31HL152686 (M.D.D.) and F31HL158212 (J.D.H.), and the NSF GRFP (M.L.K.). This work also utilized the UC San Diego IGM Genomics Center, utilizing an Illumina NovaSeq 6000 that was purchased with funding from a National Institutes of Health SIG grant (#S10 OD026929), and the UCSD School of Medicine Microscopy Core grant (P30 NS047101).

## Author contributions

J.M.M., V.K.N., K.R.K., and K.L.C. designed and performed the experiments, analyzed the data, and wrote the manuscript. M.D.D., E.G.W., R.M.W., M.L.K., J.A.P., and B.D.B. performed the experiments and analyzed the data. M.B.N., A.C., and J.D.H. performed quality control on both ECM hydrogel batches. J.Y., D.M.C., N.T., and Z.F. helped with single cell and Visium experiments. C.G.L. and R.L.B. performed rat surgeries and intramyocardial injections. K.L.C. conceived the project, designed experiments, and provided funding. All authors reviewed the results and commented on the manuscript.

## Competing interests

K.L.C. holds equity and receives income from Ventrix Bio, Inc. K.L.C. is a co-founder, consultant, and board member of Ventrix Bio, Inc. The other authors declare no competing interests related to this work.
