## [Transparent Peer Review file · Nature Communications]

Regional and Cell Specific Bioactivity of Injectable Extracellular Matrix Biomaterials in Myocardial Infarction

Corresponding Author: Dr Karen Christman

Version 0:

Reviewer comments:

Reviewer #1

(Remarks to the Author)

Mesfin et al., present an interesting article entitled "Uncovering the Regional and Cell Specific Bioactivity of Injectable ECM Biomaterials in MI through Spatial and Single Nucleus Transcriptomics." In this work, the authors aimed to unveil specific differences in cell type that take place after myocardial infarction and treatment with a porcine derived material. While the study is interesting, there are some key questions the authors need to work on and some additional comments that need of attention.

1. It is not clear how the authors define the different regions of the infarcted area, meaning infarct, border zone, and remote tissue. This is critical to adequately link the spatial transcriptome data and the experimental findings.

2. Minimum sample size calculations for spatial transcriptome are missing as well as much clearer representation for differences in cell number between the 1 and 8 weeks post-MI models.

3. The authors must report cardiac function data for all animals, as well as infarct sizes. While ischemia reperfusion is an excellent model for myocardial infarction, one must always ensure a cardiac function decline (akinetic area in echo) is present before including an animal in the experimental group.

4. Batch to batch variability must be accounted for in the experimental design. What are the quality control used for determining if the injected ECM tissue is optimal?

5. The following sentence is worrisome: "At either 1-week post-MI (subacute) or 8 weeks post-MI (chronic), animals were subjected to a 75 μ L injection of ECM hydrogel or saline with a 27G needle directly into the infarct via subxiphoid access as previously described." While understanding the intention is to deliver the therapeutic agent to the infarct zone, in much larger and established infarcts, the tissue is too frail for adequate injection, and one takes the risk of perforating the myocardium. A much more accepted protocol for intramyocardial injection is to deliver the material "around" the akinetic area using an echo-guided injection. In such protocol, the total amount to be delivered is split into 5-6 injections that covers the entire perimeter of the infarct. The authors must spend additional time explaining why they injected directly to the infarcted region.

6. The title of the article should not contain abbreviations/acronyms as those are barriers of accessibility for the non-expert in the cardiovascular field.

Reviewer #2

(Remarks to the Author)

Decellularized ECM materials have gained lots of interest in the field of tissue engineering and regenerative medicine, because of its cellular compatibility, broad applicability, together with its therapeutic efficacy. As far as I know, the team of Dr. Christman have diligently worked on ECM materials for cardiovascular application and to elucidate the underlying mechanisms of action over decades. They're now leading phase 1 clinical trial in subacute and chronic MI patients. That's great progress. Back to this manuscript, they adopted snRNA-seq coupling with spatial transcriptomics to further reveal the influence of ECM hydrogel on myocardial infarction in cellular level. The research is very inspirative and innovative to our

understanding of ECM materials for biomedical applications, and I have the following minor comments.

1. The author claimed ECM or saline was injected for subacute I/R rat treatment, yet the Visium analysis was conducted with individual sample within the same infarct regarding to regions adjacent or distal to the ECM hydrogel. This is confusing. Have the authors checked on to the zones within the ECM-treated infarct but are absent of ECM hydrogel with the infarct zones received saline injection? Is there any notable difference? Please clarify.
2. There are no quantitative ratios of different cell types in the snRNA-seq. This is critical to reveal the potential effects of ECM hydrogel for further analysis. Why do the authors determine to analysis macrophages, cardiomyocyte, fibroblast and endothelial cells over the other unclassified cell types? Is this based on the ratio changes? Are they ranked tops?
3. Again, even though there is no substantial presence of ECM hydrogel in the ECM-treated infarct 7 days later, the microenvironment can be already modified following degradation of a tiny amount of ECM hydrogel. This could be a possible situation to question the rationale of regions for analysis. This consideration goes to both the sub-acute and chronic models in the study.
4. It looks like the effects of the ECM hydrogel are scattered to act on different types of cells in the injury microenvironment. Have you found any integrated signal pathways that may rule the effects in a specific cell population?
5. How many samples have the author analyzed for each group regarding spatial transcriptomics? How consistent they are?
6. Have the authors detected any confluence of differential expressed genes between subacute and chronic samples? This can be an important advocate for the claim that the pro-repair effects is conserved across administration timepoints.

Reviewer #3

(Remarks to the Author)

This article investigates the bioactivity of injectable decellularized extracellular matrix (ECM) hydrogels in the treatment of myocardial infarction (MI). Using spatial transcriptomics and single-nucleus RNA sequencing (snRNAseq), the study reveals region-specific and cell-specific responses to ECM hydrogel administration in both subacute and chronic MI models. The results indicate that ECM hydrogel can induce pro-repair programs such as immunomodulation, vascular development, cardioprotection, and myocardial developmental programs, with consistent effects across different time points.

The article presents several critical issues that should be addressed to enhance the study's comprehensiveness and reliability. First, the interpretation of the data is insufficient; while the authors provide a substantial amount of data, the biological significance of key findings, particularly the changes in gene expression, is not thoroughly explained. This lack of detailed interpretation makes it challenging to understand the broader implications of the results. Second, the study lacks long-term investigations. Although it focuses on the subacute and chronic stages, it only repeats similar analyses and does not explore the long-term effects of ECM hydrogel treatment, which are crucial for evaluating both the durability and safety of the therapy. Third, the comparative analysis is limited. The study compares ECM hydrogel with saline, but fails to include other treatment modalities, such as stem cell therapy or pharmacological interventions, which would provide a more comprehensive assessment of the hydrogel's relative efficacy. Lastly, concerns regarding the sample size arise, as the article does not specify the number of samples for each experimental group. This omission could affect the statistical significance and reliability of the results.

Minor concerns:

1. What is the difference between Fig. 1b and Supplementary Information Fig. 1a,b? The details regarding the sample information, such as the number of spatial samples in each group (Saline and ECM hydrogel), are missing and should be clearly labeled in the figures or listed in tables. Are there only one sample for each group? Are there any replicates?
2. In Fig. 1c, how do you differentiate between the infarct-only regions and the ECM hydrogel within the infarct? Was the image aligned with adjacent sections that were labeled with fluorescence?
3. Fig. 2 is somewhat confusing and should be revised for clarity, particularly for the left and right comparison between ECM Hydrogel and Saline. These images should be presented together to better observe corresponding locations between the two groups. Additionally, there is a lack of UMAP plot annotations for the subclusters between Fig. 2a and Fig. 2b-e. Fig. 2b-e shows the subsets of each cell type and their recluster results—was this an integrated analysis? (should be integration of both condition but it is not clearly specified). The specificity of the cell clusters between the two groups appears weak, possibly indicating batch effects rather than true specificity, particularly in Fig. 2d. The same issue arises in the comparison between subacute and chronic ECM zones.
4. In line 119, how was the interaction between the ECM hydrogel and helper T-cell activation observed in Fig. 1D?
5. In line 179, I was unable to find evidence of higher cardiomyocyte proliferation within the ECM hydrogel treatment condition.
6. In line 256, there is a lack of quantitative comparison between the subacute and chronic ECM zones.
7. The manuscript lacks a detailed description of the statistical methodology used throughout.

Version 1:

Reviewer comments:

Reviewer #2

(Remarks to the Author)

The authors have nicely addressed all my critiques.

Reviewer #3

(Remarks to the Author)

The revised manuscript has shown improvement with the inclusion of additional statistical statements and detailed experimental data. However, there are several areas that require further attention.

- 1) while the snRNA-seq data was annotated using markers from other literature, the manuscript lacks a thorough presentation of data quality, feature signatures, and clear cell type labels in the UMAP plot.
- 2) again, the comparison of unique or significantly different cell populations within subclustered cells across the saline and ECM groups requires a more integrated approach. Conducting the analysis in a consensus 2D space would allow for a more comprehensive understanding of the differences and similarities between these groups, ensuring that the conclusions drawn are more robust and reliable.
- 3) the spatial data analysis, while a valuable component of the study, has yielded limited new findings, as the spatial dimension was not comprehensively explored.
- 4) the conclusions based on GO term analysis should be interpreted with caution. While GO enrichment can provide valuable insights, the manuscript must ensure that these findings are supported by affirmative experimental evidence to avoid overreaching conclusions.

Reviewer #4

(Remarks to the Author)

Thank you for the responses. Reviewer 1 comments were properly addressed.

Version 2:

Reviewer comments:

Reviewer #3

(Remarks to the Author)

The authors' provision of basic metrics for single-nucleus and spatial data is commendable, aiding in data evaluation given the inherent challenges in acquiring high-quality cardiac tissue data. For clarity, "features per pixel" refers to the number of distinct measurable characteristics (e.g., gene expression levels, protein abundance) detected within a single pixel of the spatial imaging data. Similarly, "genes per pixel" denotes the number of unique genes identified within a single pixel. Why do they differ a lot in the data both in single nucleus and spatial data? The observation of feature counts exceeding 30,000 (or 10,000 for single nucleus) in certain samples warrants further investigation to ascertain potential sources of technical artifacts or biological explanations. Finally, a detailed examination of cell proportion differences in integrated analyses (e.g., Supplementary Information Figures 5 & 8), alongside a discussion of their potential biological implications, is crucial for a comprehensive interpretation of the study's findings.

Uncovering the Regional and Cell Specific Bioactivity of Injectable Extracellular Matrix Biomaterials in Myocardial Infarction through Spatial and Single Nucleus Transcriptomics

We are thankful for the careful review of our manuscript and have revised it according to the reviewers' comments. The following is a point-by-point reply to each of the reviewers' recommendations. Our response to each point is shown in blue text after their comment in normal typeface.

REVIEWER COMMENTS

Reviewer #1 (Remarks to the Author):

Mesfin et al., present an interesting article entitled "Uncovering the Regional and Cell Specific Bioactivity of Injectable ECM Biomaterials in MI through Spatial and Single Nucleus Transcriptomics." In this work, the authors aimed to unveil specific differences in cell type that take place after myocardial infarction and treatment with a porcine derived material. While the study is interesting, there are some key questions the authors need to work on and some additional comments that need of attention.

1. It is not clear how the authors define the different regions of the infarcted area, meaning infarct, border zone, and remote tissue. This is critical to adequately link the spatial transcriptome data and the experimental findings.

We have clarified in the first paragraph of the results as well as in the section "Response to ECM hydrogel is also spatially distinct in a chronic MI model" on how we delineated the infarct zone from the border zone and remote tissue. When pan-cardiomyocyte genes are low (*Myh6* or *Tnnt2*), we call it the infarct zone. When pan-cardiomyocyte genes are high, it is either the border zone or remote zone. In this case, we only are studying the infarct zone that contains ECM hydrogel and infarct zone that does not contain ECM hydrogel. Supplementary Figures 2 and 4 now better illustrate the infarct zone for representative subacute and chronic samples.

2. Minimum sample size calculations for spatial transcriptome are missing as well as much clearer representation for differences in cell number between the 1 and 8 weeks post-MI models.

We have now explicitly defined the sample sizes for each of the spatial transcriptome samples (Table 1) alongside cellular distribution differences between the 1 and 8 week post-MI models (Figures 2B and 4B).

Here, we performed a transcriptional profiling screen, which was consistent with the biomaterial single cell (Cherry et al., Nature BME, 2021; Yang et al., Nat. Commun., 2023) and spatial transcriptomic studies (Yang et al., Nat. Commun., 2023) present in the field. In these previous studies, the samples in terms of biological n have varied from n = 1 to n = 2 per experimental group, which in total would range from 3,000 to 10,000 cells for analysis. Our study investigates the differences of anywhere from approximately 3,000 to 20,000 cells/spots per condition, a number above the amount that has been studied in previous biomaterial single cell and spatial papers.

Most importantly, we note in the discussion that a corroboration of these findings within single nucleus and spatial data allows for these techniques to gain effective power when cross-validated.

3. The authors must report cardiac function data for all animals, as well as infarct sizes. While ischemia reperfusion is an excellent model for myocardial infarction, one must always ensure a cardiac function decline (akinetetic area in echo) is present before including an animal in the experimental group.

We agree with the point regarding infarct size. We previously measured this and have now added the infarct size calculations for both spatial and snRNAseq samples at the subacute and chronic timepoints to

the paper. It is best practice in our lab to use a pre-specified cutoff for sufficient infarct size and that has been applied as with previous publications (Wassenaar et al., JACC, 2016).

We however respectively disagree regarding requiring cardiac function analysis being necessary in this context. We have over 20 years of experience with the rat ischemia reperfusion model, and showing a decline in cardiac function acutely via echo does not always equate to a sufficient infarct due to the potential for myocardial stunning. In fact, for all studies in our lab where we evaluate cardiac function as a key readout, we also follow up with confirmation via histology to ensure a sufficient infarct and set pre-specified exclusion criteria based on infarct size. Thus, in this study, we applied infarct size as our metric to exclude animals to ensure that all animals included in the current study had infarcts that were consistent with previous studies that evaluated cardiac function following treatment with the ECM hydrogel (Singelyn et al., JACC, 2009; Seif-Naraghi, Sci. Trans. Med. 2012,; Wassenaar et al., JACC, 2016; Diaz et al., JACC: BTS, 2021).

4. Batch to batch variability must be accounted for in the experimental design. What are the quality control used for determining if the injected ECM tissue is optimal?

Given that we have now translated two of our ECM hydrogels into a cGMP process, this is something we take very seriously. Our lab has generated effective quality control metrics for generating different types of decellularized ECM, which are described in a previous publication (Ungerleider et al., Methods, 2015). We rigorously evaluate each batch to ensure it meets specifications prior to use and to ensure limited batch to batch variability using similar criteria that is used for clinical grade material. We also have several standard operating procedures (SOPs) to ensure consistency.

We have now included relevant quality control data in Supplementary Information Figure 1.

5. The following sentence is worrisome: "At either 1-week post-MI (subacute) or 8 weeks post-MI (chronic), animals were subjected to a 75 μ L injection of ECM hydrogel or saline with a 27G needle directly into the infarct via subxiphoid access as previously described." While understanding the intention is to deliver the therapeutic agent to the infarct zone, in much larger and established infarcts, the tissue is too frail for adequate injection, and one takes the risk of perforating the myocardium. A much more accepted protocol for intramyocardial injection is to deliver the material "around" the akinetic area using an echo-guided injection. In such protocol, the total amount to be delivered is split into 5-6 injections that covers the entire perimeter of the infarct. The authors must spend additional time explaining why they injected directly to the infarcted region.

While it is true that an infarct can be too frail for adequate biomaterial injection into a mouse infarct (in fact this is one of the reasons we do not use mice), this is not the case with a rat infarct which is of large enough size to allow for injection of a biomaterial. We have been performing these injections for approximately 2 decades without issues of perforating the myocardium (Singelyn et al., JACC, 2009; Seif-Naraghi et al., Sci. Trans. Med. 2012; Wassenaar et al., JACC, 2016; Diaz et al., JACC: BTS, 2021). This protocol dates back >20 years to the first injectable biomaterial in a rat model of myocardial infarction (Christman et al., JACC, 2004; Christman et al., Tissue Engineering, 2004). Performing injections around the infarct/in the border zone is more common practice for cell injections, which have issues with survival; however, this is not a concern with a biomaterial only approach which can be used to directly treat and modulate the infarct. In a rat, one injection of a biomaterial spreads throughout a large portion of the infarct. In fact, we are actually more concerned with performing multiple injections in such a small heart because needle injection has also been shown to damage the myocardium and we want to limit this. The finding of needle pass injection inducing evidence of injury has also been shown through spatial transcriptomics from the King Lab (Calcagno et al., Nat. Cardiovasc. Res., 2022). In larger animals such as pigs, we have performed multiple injections (Seif-Naraghi, Sci. Trans. Med. 2012), but do not think this is the best approach for the rat and respectively disagree that multiple echo guided injections are a more accepted protocol for biomaterial injections.

6. The title of the article should not contain abbreviations/acronyms as those are barriers of accessibility for the non-expert in the cardiovascular field.

We agree. The title has been changed to reflect this.

Reviewer #2 (Remarks to the Author):

Decellularized ECM materials have gained lots of interest in the field of tissue engineering and regenerative medicine, because of its cellular compatibility, broad applicability, together with its therapeutic efficacy. As far as I know, the team of Dr. Christman have diligently worked on ECM materials for cardiovascular application and to elucidate the underlying mechanisms of action over decades. They're now leading phase 1 clinical trial in subacute and chronic MI patients. That's great progress. Back to this manuscript, they adopted snRNA-seq coupling with spatial transcriptomics to further reveal the influence of ECM hydrogel on myocardial infarction in cellular level. The research is very inspirative and innovative to our understanding of ECM materials for biomedical applications, and I have the following minor comments.

1. The author claimed ECM or saline was injected for subacute I/R rat treatment, yet the Visium analysis was conducted with individual sample within the same infarct regarding to regions adjacent or distal to the ECM hydrogel. This is confusing. Have the authors checked on to the zones within the ECM-treated infarct but are absent of ECM hydrogel with the infarct zones received saline injection? Is there any notable difference? Please clarify.

We appreciate this comment as our ECM hydrogel may not be within the entire infarct. This comparison to saline infarcts is an interesting one, and thus we integrated these samples to compare infarct areas without ECM hydrogel to saline infarcts. Extended Data Figures 1B and 5B provide slight differences with this finding, where we find that infarct zones without ECM hydrogel do have upregulation of genes found in the ECM zone relative to saline infarct zones. However, in general, there are few differentially expressed genes distinguishing the two, which signifies that there is augmented pro-repair expression in areas where the ECM hydrogel is contained.

2. There are no quantitative ratios of different cell types in the snRNA-seq. This is critical to reveal the potential effects of ECM hydrogel for further analysis. Why do the authors determine to analysis macrophages, cardiomyocyte, fibroblast and endothelial cells over the other unclassified cell types? Is this based on the ratio changes? Are they ranked tops?

We have analyzed these cell types via snRNAseq due to their prevalence in the heart (Litviňuková et al., Nature, 2020). In the manuscript, we have now studied each of the prevalent cell types that have been previously quantified in Litviňuková et al., Nature, 2020. Figures 2B and 4B now include quantitative ratios of the different cell types in the snRNAseq for the subacute and chronic studies, respectively.

In addition, we have now included comparisons for all relevant cell types that were illustrated in previous cell atlases.

3. Again, even though there is no substantial presence of ECM hydrogel in the ECM-treated infarct 7 days later, the microenvironment can be already modified following degradation of a tiny amount of ECM hydrogel. This could be a possible situation to question the rationale of regions for analysis. This consideration goes to both the sub-acute and chronic models in the study.

We acknowledge that this is a limitation of measuring the ECM hydrogel in the ECM-treated infarct 7 days later. We decided to leverage this timepoint as 7 days post injection leads to peak cell infiltration (Seif-Naraghi, Sci. Trans. Med., 2012), which would be the best timepoint to measure cellular and transcriptional changes. In addition, previous studies measuring the gene expression changes due to ECM hydrogel administration found larger gene expression differences at 7 days post-injection compared

to 3 days post-injection, lending more rationale towards the 7 day post-injection timepoint for determining the ECM's therapeutic effect per cell type and in space.

We now however acknowledge this limitation in the discussion, as it would be interesting to evaluate more acute responses.

4. It looks like the effects of the ECM hydrogel are scattered to act on different types of cells in the injury microenvironment. Have you found any integrated signal pathways that may rule the effects in a specific cell population?

We appreciate this comment, and we decided to narrow down specific pathways that are uniformly regulated with ECM administration. We thus performed ligand-receptor analysis in both the subacute [Extended Data Fig. 4] and chronic models [Extended Data Fig. 7], where we found GO terms that were found to be enriched with ligand/receptor pairings associated with ECM administration. In the subacute model [Extended Data Fig. 4], we found reduced activation of the TGF β pathway with ligand/receptor pairings associated with ECM administration, alongside blood vessel morphogenesis and T-cell activation. In the chronic model [Extended Data Fig. 7], we see higher angiogenic ligand-receptor pairings with ECM administration, while saline had higher ligand-receptor signaling involving metabolic processes.

Uniquely, we found downregulation of metabolic processes in both MI models, alongside downregulation of a particular ligand-receptor pairing: Nppa-Nrp1. This complex is canonically known to increase autophagy in the infarcted heart, particularly in cardiomyocytes, signaling that the ECM may act on this pathway to regulate autophagy due to MI.

5. How many samples have the author analyzed for each group regarding spatial transcriptomics? How consistent they are?

For spatial transcriptomics, we have n = 2 for subacute ECM hydrogel administration, n = 3 for subacute saline administration, n = 3 for chronic ECM hydrogel administration, and n = 2 chronic saline administration.

We have found that they are consistent with each other in terms of their findings, particularly with expression of pro-repair markers (immunomodulation, fibroblast activation, and vasculature development). We also found them to be corroborated with their corresponding snRNAseq data at that timepoint, demonstrating their consistency even with modest n.

6. Have the authors detected any confluence of differential expressed genes between subacute and chronic samples? This can be an important advocate for the claim that the pro-repair effects is conserved across administration timepoints.

We agree that conserved genes should be considered in claiming a conserved pro-repair response. Thus, we have added another panel in Figure 5G as a qualitative Venn-diagram comparing the subacute and chronic responses, with common responses and genes in parentheses. We note a conserved pro-repair response with fibroblast activation and vasculature development, but higher immune cell activation and smooth muscle cell proliferation in the subacute model. ECM administration in the chronic model showed higher developmental and neurogenic markers.

Reviewer #3 (Remarks to the Author):

This article investigates the bioactivity of injectable decellularized extracellular matrix (ECM) hydrogels in the treatment of myocardial infarction (MI). Using spatial transcriptomics and single-nucleus RNA sequencing (snRNAseq), the study reveals region-specific and cell-specific responses to ECM hydrogel administration in both subacute and chronic MI models. The results indicate that ECM hydrogel can

induce pro-repair programs such as immunomodulation, vascular development, cardioprotection, and myocardial developmental programs, with consistent effects across different time points.

The article presents several critical issues that should be addressed to enhance the study's comprehensiveness and reliability. First, the interpretation of the data is insufficient; while the authors provide a substantial amount of data, the biological significance of key findings, particularly the changes in gene expression, is not thoroughly explained. This lack of detailed interpretation makes it challenging to understand the broader implications of the results.

We agree that the biological significance of these key findings could be explained more. We thus provided more context towards the biological significance of these findings in the main text, especially as it refers to the comparisons between subacute and chronic MI pro-repair responses due to ECM hydrogel administration. In this case, we provided context of our findings with respect to each individual facet of pro-repair in MI mentioned in the main text (immunomodulation in MI, angiogenesis, myocardial salvage, resident macrophage salvage, lymphatic development, neurogenesis, and T-cell proliferation). Here, we demonstrate more detail on our biological findings with respect to each cell type, demonstrating evidence of pro-repair response in each cell type, alongside references as to why each is important in mitigating the effects of myocardial infarction. Finally, as mentioned in a previous response above, we also ran ligand receptor analyses to find some consensus pathways that ECM may be acting on, leading us to the downregulation of the *Nppa-Nrp1* ligand receptor pairing and thus demonstrating an effect with MI autophagy in cardiomyocytes. All of these citations and findings are now detailed in the main text.

In terms of the biological implications, we noted in previous publications that the ECM hydrogel has demonstrated therapeutic efficacy in subacute (Singelyn et al., JACC, 2009; Seif-Naraghi et al., Sci. Trans. Med. 2012; Wassenaar et al., JACC, 2016), and chronic MI (Diaz et al., JACC: BTS, 2021) from the preclinical stage to a phase 1 clinical trial (Traverse et al, JACC: BTS, 2019). However, bio-inert materials were insufficient in preventing negative left ventricular remodeling in MI (Rane et al., PLOS One, 2011). As a result, the findings in this manuscript lend evidence toward showing that ECM elicits not only a global pro-repair phenotype, but also signals specific cell populations towards pro-repair when administered at different timepoints post-MI.

Second, the study lacks long-term investigations. Although it focuses on the subacute and chronic stages, it only repeats similar analyses and does not explore the long-term effects of ECM hydrogel treatment, which are crucial for evaluating both the durability and safety of the therapy.

We appreciate this important point regarding the long-term effects of ECM hydrogel treatment. Our lab has studied the degradation kinetics of our ECM, which shows that the material degrades around 14-21 days post administration. Spatial transcriptomics evaluation of locations with and without the ECM hydrogel are not possible beyond 7 days since there is not enough hydrogel remaining. We have extensive previous work showing long term durability and safety with this therapy in both subacute and chronic models (Singelyn et al., JACC, 2009; Seif-Naraghi et al., Sci. Trans. Med. 2012; Wassenaar et al., JACC, 2016; Diaz et al., JACC: BTS, 2021), plus initial 1 year safety and feasibility in patients (Traverse et al., JACC: BTS, 2019). Thus, given that we have already have long term evaluation up to a phase 1 clinical trial, further long term evaluation was not the goal of this study.

In contrast, the goal of this work was to delve deeper into the mechanism of action of the ECM hydrogel using modern transcriptomics techniques. Initial papers on the ECM's mechanism included preliminary gene expression changes at the subacute (Wassenaar et al., JACC, 2016) and chronic timepoints (Diaz et al., JACC:BTS, 2021) via biased gene expression arrays. In this case, given that gene expression changes after the dynamic period are expected to decrease in the long term, measuring gene expression at much later timepoints would not inform much about the ECM's mechanism of action post-MI.

Third, the comparative analysis is limited. The study compares ECM hydrogel with saline, but fails to include other treatment modalities, such as stem cell therapy or pharmacological interventions, which would provide a more comprehensive assessment of the hydrogel's relative efficacy.

In this manuscript, we respectfully believe that evaluating the ECM in isolation is the best way to clarify its biological mechanisms. We agree that for translation, it will be important to evaluate the marginal benefit of therapeutic ECM on a background of guideline directed medical therapy (GDMT). Toward that end, we have pursued the ECM hydrogel in a phase 1 clinical trial, which included patients with GDMT. However, in a preclinical setting, rigorous comparison to modern 4-drug GDMT alone or in combination would require extensive comparisons and controls. From a practical perspective, clinical translation of our ECM hydrogel will undoubtedly be in conjunction with modern GDMT.

In contrast to pharmacologic GDMT, stem cell therapy for ischemic heart disease remains a frontier area of research with many variations and lack of consensus about cell source, pretreatment, number of cells, site of injection, and more, which is why this was not included as a control.

Lastly, concerns regarding the sample size arise, as the article does not specify the number of samples for each experimental group. This omission could affect the statistical significance and reliability of the results.

Regarding the spatial and single nucleus RNAseq sample number, we put this in the main text [Table 1] for easier visualization of sample size number. We also now indicate the sample size in the figure legends, alongside the number of cells/spots studied for each comparison. As mentioned in the response to reviewer 1, the sample sizes are either equivalent or larger to previous studies in the field of biomaterial scRNAseq and spatial transcriptomics. We do note this now as a limitation in the manuscript's discussion.

These type of sample sizes are more typical for scRNAseq and spatial transcriptomics due to the high cost of these studies, particularly for both sequencing and reagents, and scRNAseq aims to resolve cellular heterogeneity on a per sample basis. In this case, by keeping the harvest timepoint the same (7 days post-injection) for both subacute and chronic models of MI, alongside for all snRNAseq and spatial studies, we can utilize multiple models and transcriptomic methods to cross reference all findings between different cohorts of animals. In the manuscript, we did find that our spatial and snRNAseq data corroborated each other's findings, with some findings conserved between subacute and chronic ECM administration. Thus, keeping the timepoint consistent allowed for us to make use of small n, while comparing these findings in different sets of animals.

Minor concerns:

1. What is the difference between Fig. 1b and Supplementary Information Fig. 1a,b? The details regarding the sample information, such as the number of spatial samples in each group (Saline and ECM hydrogel), are missing and should be clearly labeled in the figures or listed in tables. Are there only one sample for each group? Are there any replicates?

To clarify, Supplementary Information Figures 2 and 4 contain how the infarct zone was defined. As mentioned in a previous answer, when pan-cardiomyocyte genes are low (*Myh6* or *Tnnt2*), we call it the infarct zone. When pan-cardiomyocyte genes are high, it is the border zone or remote zone. In this case, we only are studying the infarct zone. The distinction between Fig. 1B and Supplementary Information Fig. 2 now is clarified in the main text.

Regarding replicates, we moved the sample size table to the main text, and included the number of n in the figure legends.

2. In Fig. 1c, how do you differentiate between the infarct-only regions and the ECM hydrogel within the infarct? Was the image aligned with adjacent sections that were labeled with fluorescence?

We differentiated between the infarct-only regions and the ECM hydrogel in the infarct by aligning the spatial transcriptomic image with the immediately adjacent (next 10 μm) section that was stained with α -actinin and contained fluorescently tagged ECM hydrogel. This is now articulated in the main text.

3. Fig. 2 is somewhat confusing and should be revised for clarity, particularly for the left and right comparison between ECM Hydrogel and Saline. These images should be presented together to better observe corresponding locations between the two groups. Additionally, there is a lack of UMAP plot annotations for the subclusters between Fig. 2a and Fig. 2b-e. Fig. 2b-e shows the subsets of each cell type and their recluster results—was this an integrated analysis? (should be integration of both condition but it is not clearly specified). The specificity of the cell clusters between the two groups appears weak, possibly indicating batch effects rather than true specificity, particularly in Fig. 2d. The same issue arises in the comparison between subacute and chronic ECM zones.

We agree with the orientation of the figure, particularly the Volcano plot, to better read the overall differences. All of the differentially expressed genes relative to saline administration are on the left hand side of the Volcano Plot, and the right hand side has genes resulting from ECM hydrogel administration. The Volcano Plots across the entire manuscript are now more appropriately labeled to allow for easier viewing.

As a clarification, this was an integrated analysis of $n = 2$ samples per ECM hydrogel administration, and $n = 2$ per saline administration. This was also clarified in the main text.

4. In line 119, how was the interaction between the ECM hydrogel and helper T-cell activation observed in Fig. 1D?

The key genes that demonstrated interaction between the the ECM hydrogel and helper T-cell response are: *Lgals1*, *Pdlim3*, *Cd53*, and *Fat1*. All of which are involved in the T-cell response, particularly in an anti-inflammatory response in MI. We also have now thus clarified this in the main text.

5. In line 179, I was unable to find evidence of higher cardiomyocyte proliferation within the ECM hydrogel treatment condition.

This information is defined in the extended data section, specifically in Extended Data Figure 3. This was done by scoring cell types that were subset and separated by condition. We have now amended the figure to better contextualize scoring and on what cell types they were done on.

6. In line 256, there is a lack of quantitative comparison between the subacute and chronic ECM zones.

We think this is a useful comparison to make. We thus integrated the subacute and chronic ECM zones and thus performed a head-to-head comparison in Extended Data Figure 8, where we displayed the top differentially expressed genes within the subacute and chronic ECM zones.

7. The manuscript lacks a detailed description of the statistical methodology used throughout.

This has been adjusted in the main text's methodology:

Group size was based on previous literature of biomaterial single cell (Cherry et al., al., Nature BME, 2021; Yang et al., Nat. Commun., 2023) and spatial studies (Yang et al., Nat. Commun., 2023). Comparisons between two groups for gene list scoring were analyzed using two-tailed Mann-Whitney nonparametric tests. Differential expression analyses were performed using nonparametric Wilcoxon rank-sum tests with Benjamini–Hochberg FDR adjustment. The numerical data were presented as Violin Plots. Statistical significance was determined via $p < 0.05$, with (* $p < 0.05$, ** $p < 0.01$, *** $p < 0.001$, **** $p < 0.0001$).

The statistics utilized in this manuscript is standard practice in the field for both biomaterial transcriptomic studies alongside myocardial infarction transcriptomic studies (Cherry et al., *Nature BME*, 2021; Calcagno et al., *Nat. Cardio. Res.*, 2022; Yang et al., *Nat. Commun.* 2023; Ninh et al., *Nature*, 2024).

We are thankful for the careful review of our manuscript and have revised it according to the reviewers' comments. The following is a point-by-point reply to each of the reviewers' recommendations. Our response to each point is shown in blue text after their comment in normal typeface, with changes to the text referenced below the answer in quotations. All citations are provided.

Reviewers #1 and #2:

Please note that the original reviewer #1 was unavailable to see the manuscript this time, and was replaced by reviewer #4 who has the same expertise. Additionally, reviewer #2 confidentially commented on your replies to reviewer #1, confirming that they are satisfactory while also adding that a limitation statement regarding the lack of echo cardiac functional studies is warranted, with explanations of why they are missing.

We have now included a limitation statement regarding the lack of cardiac functional studies, and why they are missing (Lines 492-496).

Discussion, Lines 492-496:

“While our current study lacks assessment of cardiac function, we have already demonstrated benefits in cardiac function with the ECM hydrogel in both subacute [Singelyn et al., *J Am Coll Cardiology*, 2012; Wassenaar et al., *J Am Coll Cardiol*, 2016] and chronic [Diaz et al., *J Am Coll Cardiol: Basic Trans Science*, 2021] rat MI models via MRI and a subacute pig MI model via echocardiography [Seif-Naraghi et al., *Sci Trans Med*, 2013]. Here, we utilized infarct size via histology as our metric to exclude animals that did not have a sufficient infarct, which we routinely do even in studies that use cardiac imaging [Singelyn et al., *J Am Coll Cardiology*, 2012;; Wassenaar et al., *J Am Coll Cardiol*, 2016; Diaz et al., *J Am Coll Cardiol: Basic Trans Science*, 2021].”

Reviewer #3 (Remarks to the Author):

The revised manuscript has shown improvement with the inclusion of additional statistical statements and detailed experimental data. However, there are several areas that require further attention.

1) while the snRNA-seq data was annotated using markers from other literature, the manuscript lacks a thorough presentation of data quality, feature signatures, and clear cell type labels in the UMAP plot.

We agree this could be improved and have made the following additions to the paper.

We have now included data quality signatures, such as the nFeatures and nCounts per replicate in Supplementary Information Figure 3. Mitochondrial genes were removed as outlined in the Methods section, so we did not plot % mitochondrial reads.

We have included the top 5 marker genes for the subsetted cell types for both the subacute (Supplementary Information Figure 4A) and chronic models (Supplementary Information Figure 4B), demonstrating the feature signatures for the cell types studied in the manuscript. We have included the full markers for the cell types studied in the manuscript as Supplementary Tables 5 and 12.

For each subsetted cell type, we have included a feature plot demonstrating integration, coarse clustering, and a heatmap depicting the top 5 marker genes for each cluster. All cell types in the subacute model are shown in Supplementary Information Figure 5. All cell types in the chronic model are shown in Supplementary Information Figure 8.

2) again, the comparison of unique or significantly different cell populations within subclustered cells across the saline and ECM groups requires a more integrated approach. Conducting the analysis in a consensus 2D space would allow for a more comprehensive understanding of the differences and similarities between these groups, ensuring that the conclusions drawn are more robust and reliable.

We thank the reviewer for this comment. We conducted all our analyses in an integrated 2D space for all nuclei isolated from biological replicates in each condition. Subsets of specific cell types were reintegrated to share a consensus 2D space prior to differential gene expression analyses. We have detailed and edited the methods section to illustrate our process more clearly. To validate that we have done these analyses in a consensus 2D space for each subsetted cell type, we have now included feature plots depicting the cells of ECM vs. saline for each cell type in Supplementary Information Figures 5 and 8 (see at end of Point by Point).

1) Methods, Lines 599-643:

Quality Control, Normalization, and Integration

All snRNAseq and Visium data analysis was done using the Seurat package (v4) in R. Both sets of data had raw counts for each gene normalized to specific transcript count and log-transformed. Ribosomal and mitochondrial genes were filtered out as done previously [Calcagno et al., *Nat Cardiovasc Res*, 2022], and cells/pixels with greater than 200 counts were retained for further analysis. Doublets were removed by determining cells that contained non-endogenous gene markers as done previously [Calcagno et al., *Nat Cardiovasc Res*, 2022]. Highly variable genes across individual samples were determined via the FindVariableFeatures method from Seurat R package v4 to find the top 2,000 genes with the highest feature variance.

For analyses between ECM hydrogel and saline treatment groups in snRNAseq, integration of all snRNAseq replicates was performed in Seurat to enable harmonized clustering and downstream comparative analyses [Calcagno et al., *Nat Cardiovasc Res*, 2022]. Canonical correlation analysis (CCA) was used to determine anchoring cell pairs, and integration anchors were detected through the FindIntegrationAnchors function, utilizing reference-based integration of the single nucleus datasets via CCA. Once the ECM and saline datasets were integrated together, the integrated set was then subjected to principal component analysis, and further reduced through uniform manifold approximation and projection (UMAP) to subject the data into a consensus UMAP space. The data was then clustered and visualized in UMAP space. All canonical cell types (macrophages, endothelial cells, cardiomyocytes, fibroblasts, T-cells, neuronal cells, smooth muscle cells, and lymphatic endothelial cells) were identified through literature review of the gene signatures [Litviňuková et al., *Nature*, 2020], alongside using the FindAllMarkers function in Seurat v4 ($P_{adj} < 0.05$, $\logFC > 0.25$, $\min.diff.pct > 0.25$, assay = RNA) [Supplementary Information Figure 4, Supplementary Table 5, Supplementary Table 12]. Spatial transcriptomic analyses of ECM and saline treatment groups were subjected to the same integration process to ensure comparisons were done in a consensus UMAP space.

For each relevant cell type, the data were subsetted through novel gene markers, integrated via CCA, renormalized, subjected to principal component analysis, and reduced through UMAP to subject each cell type across treatment condition into a consensus UMAP space. Integration for each cell type in the subacute model is depicted in Supplementary Information Figure 5. Integration for each cell type in the chronic model is depicted in Supplementary Information Figure 8.

Coarse Clustering and Differential Gene Expression Analyses

snRNAseq data were integrated as described above and clustered (resolution = 1). As mentioned above, each canonical cell type (macrophages, endothelial cells, cardiomyocytes, fibroblasts, T-cells, neuronal cells, smooth muscle cells, and lymphatic endothelial cells) was subsetted regardless of condition into a consensus 2D UMAP space. These subsets per cell type were subjected to further coarse clustering (resolution = 1). Subset-specific features per cluster lists were determined using FindAllMarkers ($P_{\text{adj}} < 0.05$, $\log\text{FC} > 0.25$, $\text{min.diff.pct} > 0.25$), displayed as a heatmap with the top 5 features per cluster, and are displayed in Supplementary Table 6 for the subacute model, and Supplementary Table 13 for the chronic model. Visium data were integrated as described above and clustered (resolution = 1).

All analyses between ECM and saline were conducted using the FindMarkers function in Seurat v4 ($P_{\text{adj}} < 0.05$, $\log\text{FC} > 0.25$, $\text{min.diff.pct} > 0.25$, $\text{assay} = \text{RNA} [\text{snRNAseq}]$ or $\text{assay} = \text{Spatial} [\text{Visium}]$).

3) the spatial data analysis, while a valuable component of the study, has yielded limited new findings, as the spatial dimension was not comprehensively explored.

We do agree that we can further utilize spatial transcriptomics to answer more questions about regional ECM bioactivity. While it is known that ECM can induce effects in remote areas, we wanted to further understand this phenomenon in MI. We thus compared the remote zone between ECM and saline administration. We integrated our ECM and saline datasets and compared the two in a consensus 2D UMAP space. We have now included further analyses of the remote zones of the heart between ECM and saline treatment for the subacute model (Extended Data Figure 1C) and the chronic model (Extended Data Figure 5C). We also compared the remote zones when comparing ECM administration between subacute and chronic models of MI (Extended Data Figure 8C).

We found that the ECM also exhibited increased reparative and anti-inflammatory genes patterns (Lines 110-115, Lines 262-270, Lines 395-399), alongside reducing apoptotic gene expression patterns in subacute and chronic MI models.

1) Results, Lines 110-115:

“In addition to evaluating the regional effects of ECM hydrogel administration, we compared the remote zones of ECM hydrogel treated hearts to ones treated with saline. Here, the ECM hydrogel elicited markers for the anti-inflammatory response (*Angptl4*, *Gata3*, *Bcl11b*, *Il1r2*), while saline exhibited higher pro-inflammation markers (*Ccl2*, *Ccl3*, *Ccl7*, *Csf1*, *Ccl19*, *Il17ra*), and apoptotic processes (*Spn*, *Pdpn*, *Casp12*, *Inhba*). These comparisons are displayed in Supplementary Table 4.”

2) Results, Lines 262-270:

“Finally, we compared the remote zones in ECM hydrogel treated hearts to those in the saline group in the chronic model, where we found genes involved in muscle development (*Cyp26b1*, *Col11a1*, *Col5a2*, *Tnn*) and cell cycle processes (*Cdc20*, *Ccne1*, *Ncaph*, *Smc2*) in the remote zone with ECM hydrogel treatment [Extended Data Fig. 5C]. In comparison, the saline remote zone exhibited significant inflammatory genes (*Ccl24*, *Ccl1*, *Tlr2*) and apoptotic processes (*Spn*, *Ptpn6*, *Il27ra*, *Pak1*), demonstrating that the ECM hydrogel also has significant effects in mitigating inflammation and apoptosis in the remote zone of the heart at chronic timepoints. For these comparisons, all differentially expressed genes are displayed in Supplementary Table 11.”

3) Discussion, Lines 395-399:

“Building upon other work in the field [Sadtler et al., *Science*, 2016] that elucidated the remote effects of another ECM biomaterial treatment, we utilized spatial transcriptomics to further elucidate regional bioactivity in the remote zone of the heart. Thus, we were able to

demonstrated anti-inflammatory effects in the remote region of ECM treated hearts, alongside less pro-inflammation and apoptotic effects.”

When comparing the subacute and chronic remote zones with ECM hydrogel administration, we noted less differentially expressed genes in the chronic model (Lines 347-350, Lines 419-422).

1) Results, Lines 347-350:

“With samples treated with ECM hydrogel, we also compared the subacute and chronic remote zones. Here, we found higher inflammatory processes (*Gata3*, *Lilrb4*) and development (*Vegfd*, *Pgf*, *Ccn3*, *Serpine1*) in the subacute model, with less differentially expressed genes in the chronic model [Extended Data Fig. 8C]. These findings are displayed in Supplementary Table 17.”

2) Discussion, Lines 436-439:

“Through spatial transcriptomics, we also demonstrated how the ECM hydrogel can elicit cues in the remote myocardium when comparing between the subacute and chronic timepoints, demonstrating that the ECM hydrogel can elicit pro-repair responses in remote areas of the heart.”

4) the conclusions based on GO term analysis should be interpreted with caution. While GO enrichment can provide valuable insights, the manuscript must ensure that these findings are supported by affirmative experimental evidence to avoid overreaching conclusions.

We agree and have added the following statement:

Discussion, Lines 471-475:

“While we note that GO analyses should be interpreted with caution, the GO term results are consistent with previously published findings including neovascularization [Singelyn et al., *Biomaterials*, 2009; Wassenaar et al., *J Am Coll Cardiol*, 2016], immunomodulatory processes [Wang et al. *Biomaterials*, 2017; Diaz et al., *J Am Coll Cardiol: Basic Trans Science*, 2021], fibroblast mediated responses [Ungerleider et al., *J Am Coll Cardiol: Basic Trans Science*, 2016], myocardial salvage and developmental gene activation [Singelyn et al., *Biomaterials*, 2009; Singelyn et al., *J Am Coll Cardiol: Basic Trans Science*, 2012; Wassenaar et al., *J Am Coll Cardiol*, 2016], and smooth muscle cell proliferation [Ungerleider et al., *J Am Coll Cardiol: Basic Trans Science*, 2016].”

We note that this manuscript is the first to offer evidence of an ECM biomaterial’s therapeutic benefit in enhancing lymphatic development and neurogenesis in tissue (ECM hydrogels have been used for peripheral nerve regeneration, but have not been studied for neurogenesis in other tissues), two findings that provide frameworks for future manuscripts. We have thus acknowledged this limitation:

Discussion, Lines 475-478:

“We also found that ECM hydrogels promote lymphatic endothelial cell development and neurogenesis. While a limitation of this study is the lack of other analyses on these cell types, these transcriptomic studies prompt future studies on how ECM biomaterials impact lymphangiogenesis and neurogenesis in tissue.”

Supplementary Information Figure 3. Transcriptomic quality control for Visium and snRNAseq samples. (a) Quality metrics of samples and replicates for Visium samples represented in features per pixel (nFeatures) and genes per sample (nCounts). Data are presented as box and whisker plots. **(b)** Quality metrics of samples and replicates for snRNAseq samples represented in features per nuclei (nFeatures) and genes per nuclei (nCounts). Data are presented as box and whisker plots.

a

b

Supplementary Information Figure 4. Feature subsetting and top genes for canonical cell types. (a) Gene subsetting strategy overlaid onto UMAP for subacute cell types, alongside heatmap of top 5 genes per each cell type. **(b)** Gene subsetting strategy overlaid onto UMAP for chronic cell types, alongside heatmap of top 5 genes per each cell type.

2 **Supplementary Information Figure 5. Subclusters of subsetted cell types in subacute MI.**

3 **(a)** Macrophages, **(b)** T-cells, **(c)** endothelial cells, **(d)** lymphatic endothelial cells, **(e)**
4 cardiomyocytes, **(f)** fibroblasts, **(g)** neuronal cells, and **(h)** smooth muscle cells were subsetted
5 and reintegrated. They were then reclustered at resolution 1, with the top 5 marker genes for each
6 subcluster.

7

8

9

10

11

14
15
16
17
18
19
20
21
22
23
24
25
26
27
28
29
30
31
32
33
34
35
36
37
38
39

Supplementary Information Figure 8. Subclusters of subsetted cell types in chronic MI.

(a) Macrophages, (b) endothelial cells, (c) lymphatic endothelial cells, (d) cardiomyocytes, (e) neuronal cells, (f) fibroblasts, and (g) smooth muscle cells were subsetted and reintegrated. They were then reclustered at resolution 1, with the top 5 marker genes for each subcluster.

40
41
42
43
44
45
46
47
48
49

Extended Data Figure 1. Further Subacute Visium Phenotypic Differences between ECM Hydrogel and Saline Treatment. (a) Top differentially expressed genes for the subacute MI model. (b) Comparison of infarcts between conditions, where spatial transcriptomic data of ECM hydrogel treated infarcts without fluorescent ECM is compared to saline infarcts. (c) Top differentially expressed genes for the remote zones of ECM hydrogel treated spatial samples vs. the remote zones of saline treated samples. Sample size: $n = 2$ subacute ECM hydrogel (7658 spots), $n = 3$ subacute saline (8036 spots)

Extended Data Figure 5. Further chronic Visium phenotypic differences between ECM

hydrogel and saline treatment. (a) Top differentially expressed genes for the chronic MI

model. **(b)** Comparison of infarcts between conditions, where spatial transcriptomic data of ECM

hydrogel treated infarcts without fluorescent ECM is compared to saline infarcts. **(c)** Top

differentially expressed genes for the remote zones of ECM hydrogel treated spatial samples vs.

the remote zones of saline treated samples. Sample size: $n = 3$ ECM hydrogel replicates, 9594

spots; $n = 2$ saline replicates, 8166 spots.

Extended Data Figure 8. Direct Comparison of ECM Hydrogel Administration in Subacute and Chronic Models of MI. (a) Spatial comparison of integrated subacute and chronic MI ECM zones with infarct only zones. The matrix zone's differentially expressed genes were subjected to GO enrichment. **(b)** Direct comparison of spatial subacute and chronic ECM zones, with differentially expressed genes displayed via Volcano Plot. GO enrichment related to the subacute ECM zone are also displayed. **(c)** Top differentially expressed genes for the remote zones of subacute ECM hydrogel treated spatial samples vs. the remote zones of chronic ECM hydrogel treated samples. Sample size: n = 2 subacute ECM hydrogel (7658 spots); n = 3 chronic ECM hydrogel (9594 spots).

We are thankful for the careful review of our manuscript and have revised it according to the reviewers' comments. The following is a point-by-point reply to each of the reviewers' recommendations. Our response to each point is shown in blue text after their comment in normal typeface, with changes to the text referenced below the answer in quotations. All citations are provided.

Reviewer #3 (Remarks to the Author):

The authors' provision of basic metrics for single-nucleus and spatial data is commendable, aiding in data evaluation given the inherent challenges in acquiring high-quality cardiac tissue data. For clarity, "features per pixel" refers to the number of distinct measurable characteristics (e.g., gene expression levels, protein abundance) detected within a single pixel of the spatial imaging data. Similarly, "genes per pixel" denotes the number of unique genes identified within a single pixel. Why do they differ a lot in the data both in single nucleus and spatial data? The observation of feature counts exceeding 30,000 (or 10,000 for single nucleus) in certain samples warrants further investigation to ascertain potential sources of technical artifacts or biological explanations.

We appreciate the careful review. Indeed, there was an error in our plotting. We have now corrected it and the numbers make more sense (see attached). Of note, in box and whisker plots, by convention, the bar extends up to most extreme data point. After correcting the error, the extreme is now ~1 order of magnitude lower than before. We thank Reviewer 3 for catching this. Please see the new Supplementary Information Figure 3 below.

Finally, a detailed examination of cell proportion differences in integrated analyses (e.g., Supplementary Information Figures 5 & 8), alongside a discussion of their potential biological implications, is crucial for a comprehensive interpretation of the study's findings.

This is a good suggestion and we have added cell proportions as stacked bar plots (see attached), similar to our prior publications (Calcagno et al., Nature CVR 2022 and Ninh et al., Nature 2024). It does highlight differences, but they are things already discussed in the primary manuscript. They are now added to the supplementary information section as Supplementary Information Figures 7 and 15.

Supplementary Information Figure 3. Transcriptomic quality control for Visium and snRNAseq samples. (a) Quality metrics of samples and replicates for Visium samples represented in features per pixel (nFeatures) and genes per sample (nCounts). Data are presented as box and whisker plots. **(b)** Quality metrics of samples and replicates for snRNAseq samples represented in features per nuclei (nFeatures) and genes per nuclei (nCounts). Data are presented as box and whisker plots.

Supplementary Information Figure 7. Cell proportions by treatment of subsetted cell types in subacute myocardial infarction. (a-h) Average subcluster composition of ECM hydrogel and saline for subsetted macrophages (a), T-cells (b), endothelial cells (c), lymphatic endothelial cells (d), cardiomyocytes (e), fibroblasts (f), neuronal cells (g), and smooth muscle cells (h).

Supplementary Information Figure 15. Cell Proportions by treatment of subsetting cell types in chronic myocardial infarction. (a-h) Average subcluster composition of ECM hydrogel and saline for subsetting macrophages (a), endothelial cells (b), lymphatic endothelial cells (c), cardiomyocytes (d), neuronal cells (e), fibroblasts (f), and smooth muscle cells (g).